



# A Late Quaternary climate record based on long chain diol proxies from the Chilean margin

Marijke W. de Bar[1], Dave J. Stolwijk[1,2], Jerry F. McManus[3], Jaap S. Sinninghe Damsté[1,2], Stefan Schouten[1,2]

[1]Department of Marine Microbiology and Biogeochemistry, NIOZ Royal Netherlands Institute for Sea Research, and Utrecht University, Den Burg, 1790 AB, Texel, the Netherlands
[2]Department of Earth Sciences, Faculty of Geosciences, Utrecht University, Utrecht, 3584 CB, the Netherlands
[3]Lamont-Doherty Earth Observatory, Columbia University, New York, 10964, USA

*Correspondence to*: Marijke W. de Bar (marijke.de.bar@nioz.nl)

**Abstract.** The primary focus of this study is to test the applicability of different paleoenvironmental proxies based on long chain diols, i.e., the LDI as proxy for past SST, the Diol Index as indicator of past upwelling conditions and the NDI as quantitative proxy for nitrate and phosphate concentrations in seawater. The proxies were analyzed in marine sediments recovered at ODP Site 1234, located within the Peru-Chile upwelling system, with a ~2 kyr resolution, covering the last 150 kyrs, i.e., encompassing several glacial and interglacial periods. We also generated $TEX^H_{86}$ and $U^{K'}_{37}$ temperature and planktonic $\delta^{18}O$ records, as well as TOC and accumulation rates (ARs) of TOC and lipid biomarkers (i.e., $C_{37}$ alkenones, GDGTs, dinosterol and loliolide) to reconstruct past phytoplankton production. The LDI-derived SST record co-varies with $TEX^H_{86}$- and $U^{K'}_{37}$-derived SST records as well as with the planktonic $\delta^{18}O$ record, implying that the LDI reflects past SST variations at this site. TOC and phytoplankton AR records indicate increased export production during the Last Interglacial (MIS 5), simultaneous with a peak in the abundance of preserved *Chaetoceros* diatoms, suggesting intensified upwelling during this period. The Diol Index is relatively low during the upwelling period, but peaks before and after this period, suggesting that *Proboscia* diatoms were more dominant before and after the period of upwelling. The NDI reveals the same variations as the Diol Index suggesting that the input of nitrate and phosphate was minimal during upwelling, which is unrealistic. We suggest that the Diol Index should perhaps be considered as an indicator for *Proboscia* (multiple species) productivity instead of upwelling per se, whereas the NDI likely reflects *Proboscia alata* productivity, and might therefore not be suitable as a more general paleonutrient proxy.

## 1 Introduction

Paleoclimatic reconstructions typically rely on physical, biological, and geochemical proxies from sedimentary archives. Physical proxies include sediment composition, structure, grain size, density, and magnetic susceptibility, which may provide information on the paleodepositional environment. Biological proxies comprise preserved (micro-)organisms such as diatoms, foraminifera, dinoflagellates, corals, and mollusks, or remnants deriving from higher plants such as pollen and spores.



Geochemical proxies are based on the chemical composition of either the sediment or fossilized organisms. Stable isotope and elemental concentrations are inorganic proxies typically measured in shells or skeletons of marine organisms (foraminifera, mollusks, corals) preserved in the sediment, providing insight into the chemistry of the seawater in which the organisms lived. Organic proxies are a relatively newer class of tools, based on fossilized molecules that are unique for a specific organism or

group of organisms, referred to as biomarkers. The ratios of specific biomarker molecules are often related to physical parameters of the environment in which the source organism grew, such as temperature, salinity, oxygen availability or productivity, and therefore such organic proxies can also be applied to reconstruct past depositional environments (e.g., Brassell et al., 1986; Prahl and Wakeham, 1987; Schouten et al., 2002; Rampen et al., 2008, 2012; Willmott et al., 2010; Gal et al., 2018).

In the last decade long chain diols (LCDs) have attracted attention as novel proxies to reconstruct past environmental conditions. The Long chain Diol Index (LDI) was proposed, based on the distribution of $C_{28}$ and $C_{30}$ 1,13- and 1,15-diols in marine surface sediments, which shows a good correlation with annual mean sea surface temperature (SST; Rampen et al., 2012). These compounds have been detected in cultures of Eustigmatophyte algae (e.g., Volkman et al., 1992; 1999; Méjanelle et al., 2003; Rampen et al., 2014a), but the LCD distributions observed are different to those observed in the marine realm

(e.g., Versteegh et al., 1997; 2000; Rampen et al., 2012; 2014a), and hence, their role as LCD producers in the ocean remains uncertain. The Diol Index, which is an indicator for past upwelling/high nutrient conditions, is defined as the ratio of 1,14-diols versus 1,13-diols (Willmott et al., 2010) or the $C_{30}$ 1,15-diol (Rampen et al., 2008), where 1,14-diols are biomarkers for *Proboscia* diatoms (Sinninghe Damsté et al., 2003; Rampen et al., 2014b). *Proboscia* diatoms grow in the early stages of upwelling when nutrients strongly increase in concentration (Koning et al., 2001) and, therefore, it was proposed that the

relative abundance of 1,14-diols can trace past upwelling conditions. $C_{28}$, $C_{30}$ and $C_{32}$ 1,14-diols have also been observed in the marine Dictyochophyte *Apedinella radians* (Rampen et al., 2011) but its role as 1,14-diol producer in the marine realm is unknown. Recently, a new index based on the saturated and mono-unsaturated $C_{28}$ 1,14-diol was proposed as a quantitative proxy for nitrate and phosphate concentrations (Gal et al., 2018), called the nutrient diol index (NDI). The authors found a strong positive correlation between the NDI and phosphate ($R^2 = 0.85$) and nitrate ($R^2 = 0.80$) concentrations for the sea surface

sediment data sets of Rampen et al. (2014b) and de Bar et al. (2016). This suggests that the NDI might be a good indicator of past nutrient variations in surface waters.

Preliminary applications in sediment cores of these LCD proxies have shown that the LDI, as well as the Diol Index, are promising as paleotemperature and past upwelling proxies (e.g., Pancost et al., 2009; Lopes dos Santos et al., 2012; Rampen et al., 2012; Seki et al., 2012; Rodrigo-Gámiz et al., 2014; Plancq et al., 2015; Jonas et al., 2017). However, a number of

uncertainties still exist in the application of these biomarkers. For example, recent studies of surface sediments from coastal regions reveal different 1,13- and 1,15-diol distributions compared to open ocean sediments (de Bar et al., 2016; Lattaud et al., 2017a; 2017b). Relatively high fractional abundances of the $C_{32}$ 1,15-diol along the coast were observed, as a result of riverine input, significantly affecting the LDI signal, likely due to different 1,13- and/or 1,15-diol producers thriving in river outflow waters. Moreover, studies have related high *Proboscia* diatom abundances to stratified instead of upwelling conditions (e.g.,

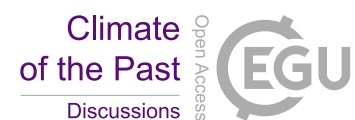

Fernández and Bode, 1994). Similarly, Contreras et al. (2010) determined the relative abundance of the $C_{28}$ 1,14-diol at ODP Site 1229 located in the Peruvian upwelling system, and observed increased concentrations during the Last Interglacial (LIG; MIS 5e) and related this to enhanced water column stratification. Furthermore, Rodrigo-Gámiz et al. (2015) showed that for sediment traps and surface sediments around Iceland, which were characterized by high concentrations of 1,14 diols (>75% of

all LCDs), the LDI did not correspond to SST. This makes application of the LDI at sites with high input of diols from *Proboscia* diatoms (e.g., upwelling sites) uncertain. Finally, due to its recent development no studies have been performed yet to test the applicability of the NDI as a paleo-nutrient proxy.

To constrain the uncertainties in applying the LCD proxies (LDI, Diol Index and NDI) at sites with upwelling and riverine

input, we studied the late Quaternary and Holocene sedimentary record at ODP Site 1234. This site is located along the Chilean margin, within the Chile-Peruvian upwelling system, and near two major river mouths of the Andean river systems Río Bio-Bio and Río Itata, both draining large basins (Muratli et al., 2010a). We sampled the last ~150 kyrs, covering the last large climate cycle of the Pleistocene, including several glacial and interglacial periods, and generated LDI, $TEX_{86}$ and $U^{K'}_{37}$ based temperature records to constrain glacial-interglacial variations in SST throughout this entire interval. Additionally, we

compared the Diol Index and the NDI record with other paleoproductivity indicators, including bulk organic matter total organic carbon (TOC), organic matter stable carbon isotopes ($\delta^{13}C$), as well as phytoplanktonic lipid biomarkers that are characteristic for certain phytoplankton communities ($C_{37}$ alkenones, loliolide and dinosterol).

## 2 Materials and methods

### 2.1 Study site

Marine sediments along the coast of Chile and Peru have been thoroughly studied, as it is a key region in the southern hemisphere for studying climate variability related to both atmospheric and oceanographic circulation (e.g., Lamy et al., 1998, 1999, 2002, 2004; Hebbeln et al., 2000, 2002; Mohtadi & Hebbeln, 2004; Stuut and Lamy, 2004; Heusser et al., 2006; Romero et al., 2006; Mohtadi et al., 2008; Muratli et al., 2010a, 2010b; Verleye & Louwye, 2010;Chase et al., 2014). The main

circulation patterns include the Southern Westerly Winds (SWW) and the Antarctic Circumpolar Current (ACC; Fig 1). The ACC approaches the South American continent from the west, and initiates both the Humboldt Current (or Peru Chile Current) flowing northwards along the continental margin and the Cape Horn Current flowing southwards (e.g., Stuut et al., 2006; Fig. 1). The Humboldt Current flows along the Chilean coast and turns westwards as it approaches the equator, forming the South Equatorial Current (SEC). Variations in strength and location of the ACC and Southern Westerlies are thought to be the main

climate controls in this region. However, the continental margin is also under influence of terrestrial input due to rainfall and land erosion (Lamy et al., 2004), and terrestrial material deposited along the coastal margin derive from mainly two mountain ranges, i.e., the Coast Range and the Andes (Lopez-Escobar et al., 1977; Martin et al., 1999).

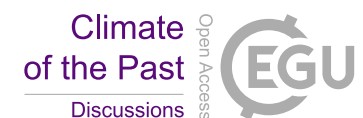

ODP Site 1234 is located in the south east Pacific Ocean (36°13.153´S,73°40.902´W, 1015 m water depth), ca. 65 km offshore of Concepción, Chile. This site lies in the vicinity of two large Andean river systems: the Río Bio-Bio and Río Itata, which both drain large basins of 24,000 km² and 11,200 km², respectively (Muratli et al., 2010a). Moreover, ODP Site 1234 is located within the Peru-Chile upwelling system, which is one of the most important upwelling regions on Earth (Berger et al., 1987),

due to the combination of upwelling-favorable winds (Daneri et al., 2000; Carr and Kearns, 2003) and an eastern boundary current (Humboldt Current). The upwelling regime stretches from 5 °S to 38 °S, corresponding to ca. 5000 km of coastline. South of 38 °S, onshore winds suppress upwelling (Strub et al., 1998),  but high productivity is sustained by relatively high nutrient supply from the ACC (Hebbeln et al., 2000), which is typically rich in nitrate and limited in micronutrients such as iron (de Baar et al., 1995). Along the Chilean coast, iron availability (as well as other micronutrients) can increase due to

fluvial supply as the precipitation of the Southern Westerlies onshore enhances river runoff (Mohtadi & Hebbeln, 2004).

### 2.2 Age model and sample strategy

The core was drilled as part of Ocean Drilling Program (ODP) Leg 202. Sedimentation rates were relatively high (~80 cm/kyr on average), providing the potential for high-resolution records (Mix et al., 2003; Heusser et al., 2006). The age model used is from Heusser et al. (2006) and is based upon radiocarbon dates and correlation of benthic $\delta^{18}$O to deep Atlantic core MD95204

(upper half) and the Vostok ice core chronology (lower half; see Heusser et al., 2006 and references therein). Heusser et al. (2006) switched between the two age models around 69 ka where they overlap, as indicated by the dashed line in Fig. 2a. However, this results in a substantial, and likely unrealistic, dip in the sedimentation rate between ca. 80 and 90 ka in the hiatus between the age models (Fig. 2b). Hence, we chose to use the upper age model (dark blue in Fig. 2) which extends up to ~83 ka, and linearly interpolate between this age and the age of ~90 ka constrained by the other age model for the lower half of the

core. This depth-age range of linear interpolation between the two age models is highlighted in green in Fig. 2 and results in less abrupt changes in sedimentation rates. The core was sampled with a ~2 kyr resolution, covering the last ~150 kyrs. In total, 74 sediment samples were analyzed for bulk and organic geochemistry.

### 2.3 Elemental analysis

All 74 sediment samples were freeze-dried and homogenized. Small aliquots (ca. 50 – 100 mg) were used for elemental analysis. For this purpose, all aliquots were acidified with 2 M hydrochloric acid (HCl) to remove all carbonates, and rinsed with distilled water to remove salts. Subsequently, the decalcified sediment samples were analyzed for total organic carbon (TOC), total nitrogen (TN) and stable carbon isotope ratios ($\delta^{13}$C) by means of a Thermo Scientific Flash 2000 Elemental Analyzer coupled to a Thermo Scientific Delta V Advantage Isotope Ratio Mass Spectrometer. Results are expressed in

standard δ-notation relative to Vienna Pee Dee Belemnite (VPDB) for $\delta^{13}$C. The precision as determined using laboratory standards calibrated to certified international reference standards, were in all cases <0.2 ‰. TOC mass ARs (MAR$_{TOC}$) were



calculated by multiplication of the sedimentation rate (linear interpolation between depth points of the age-depth model; see above) with an estimated bulk density of 1.6 g cm$^{-3}$ (Mix et al., 2003) and subsequent multiplication with the TOC percentage.

## 2.4 Foraminiferal stable isotope analysis

Benthic stable isotope data were previously published, and were generated using standard techniques in laboratories at Oregon State University and the Woods Hole Oceanographic Institution (McManus et al., 1999, 2002, 2003; Heusser et al., 2006). Oxygen isotope ratios in *Cibicidoides* and *Uvigerina* were adjusted to each other by 0.64 ‰ (Shackleton, 1974). Additional data on the planktonic species *Globigerina bulloides* were generated at Lamont Doherty Earth Observatory (LDEO) of Columbia University using a Thermo Delta V Plus gas-source isotope ratio mass spectrometer (IRMS) equipped with a Kiel

IV automated individual acid bath sample-preparation device. Although a recent study found no size-related influence on *G. bulloides* $\delta^{18}$O (Costa et al., 2017) specimens were generally picked from the 250–300 μm size range, with 8-12 individuals selected for analysis. Samples were replicated at ~5–10% frequency and measured isotope ratios were calibrated to the VPDB isotope scale with NBS-19 and NBS-18. Reproducibility of the in-house standard (1 sigma) is ±0.06‰ ($\delta^{18}$O) and ±0.04‰ ($\delta^{13}$C).

## 2.5 Lipid extraction and organic geochemical analysis

The sediment samples (ca. 15 g dry weight) were extracted using accelerated solvent extraction (ASE) using a DIONEX 200, at 100 °C, a pressure of 7–8*10$^6$ Pa, and a mixture of dichloromethane (DCM) and methanol (MeOH) (9:1; v:v). The total lipid extracts (TLEs) were dried under a stream of nitrogen (N$_2$) using a Caliper TurboVap LV. All TLEs were desulfurized

using copper granules activated with 1M HCL. The copper turnings were added to the TLEs and stirred overnight and subsequently dried over anhydrous sodium sulfate (Na$_2$SO$_4$), in order to remove precipitate and water, and dried down under N$_2$. For quantification purposes, three internal standards were added to the TLEs: 10-nonadecanone (C$_{19:0}$ ketone) for long chain alkenones, C$_{22}$ 7,16-diol for LCDs (Rodrigo-Gamiz et al., 2015) and the C$_{46}$ GDGT for GDGTs (Huguet et al., 2006). The TLEs (aliquots of ~4.5 mg) were separated into apolar, ketone (containing alkenones) and polar (containing LCDs and

GDGTs) fractions, by separation over activated (2h at 150 °C) Al$_2$O$_3$ and elution with hexane/DCM (9:1; v:v), hexane/DCM (1:1; v:v) and DCM/MeOH (1:1; v:v), respectively. Polar fractions were split for GDGT (25%) and LCD (75%) analysis.

### 2.5.1 GDGTs

Aliquots of the polar fractions were dissolved in hexane:isopropanol (99:1, v:v) to a concentration of ca. 2 mg mL$^{-1}$. The

fractions were then filtered through 0.45 μm polytetrafluoroethylene (PTFE) filters. GDGTs were analyzed by means of Ultra High Performance Liquid Chromatography Mass Spectrometry (UHPLC-MS), on an Agilent 1260 HPLC, equipped with



automatic injector, coupled to a 6130 Agilent MSD and HP Chemstation software according to Hopmans et al. (2016). The injection volume was 10 µL. Separation of the GDGTs was achieved in normal phase using 2 silica BEH HILIC columns in series (150 mm x 2.1 mm; 1,7 µm; Waters Acquity) at a temperature of 25 °C. The mobile phases are hexane (A) and hexane:isopropanol (9:1, v:v) (B). Compounds were isocratically eluted for 25 minutes with 18% B, followed by a linear

gradient to 35% B in 25 minutes and a linear gradient to 100% B in the 30 min thereafter. The flow rate was kept constant (0.2 mL/min) during the analysis. The conditions for the APCI-MS were identical to Hopmans et al. (2016). GDGTs were detected in single ion monitoring (SIM) mode of the protonated molecules ($[M+H]^+$) of the various GDGTs. A standard mixture of $C_{46}$ GDGT (internal standard) and crenarchaeol was analyzed to determine the relative response factor (RFF) between these two compounds (c.f. Huguet et al., 2006), and thereby quantify GDGTs in the sediments.

For reconstruction of past SST we used the $TEX^H_{86}$ index as proposed by Kim et al. (2010), which is defined as the logarithmic function of the original $TEX_{86}$ (Schouten et al., 2002):

$$TEX^H_{86} = \log \frac{[GDGT-2] + [GDGT-3] + [Cren']}{[GDGT-1] + [GDGT-2] + [GDGT-3] + [Cren']} \qquad (1)$$

where the numbers correspond to the amount of cyclopentane moieties in the isoprenoid GDGTs and where Cren´ refers to the later eluting isomer of crenarchaeol (Sinninghe Damsté et al., 2002). We have discarded 13 samples for the $TEX_{86}$ calculation

due to the partial co-elution of GDGT-2 with an unknown compound.

The $TEX^H_{86}$ values were converted to SSTs applying the global core top calibration of Kim et al. (2010):

$$SST = 68.4 \times TEX^H_{86} + 38.6 \qquad (2)$$

There is also a regional $TEX^H_{86}$ calibration available, based on Chilean surface sediments between 25 and 50 °S (Kaiser et al.,

20 2015):

$$SST = 59.6 \times TEX^H_{86} + 33.0 \qquad (3)$$

This calibration has a similar slope compared to the global core-top calibration of Kim et al. (2010) but a lower intercept, which results in SSTs that were ca. 2.4 °C lower compared to the $TEX^H_{86}$-SSTs calculated after Kim et al. (2010). Although the outcomes are relatively similar, the calibration of Kim et al. (2010) resulted in SSTs that agreed better with the $U^{K'}_{37}$ and

LDI records, and therefore we have used this calibration. In Appendix A, the $TEX^H_{86}$ temperatures based on the calibration of Kaiser et al. (2015) are plotted (Fig. A1). Additionally, we plotted $TEX_{86}$ temperatures calculated after the Bayesian calibration of Tierney and Tingley (2014, 2015) in Fig S1. These SSTs are on average 4 °C lower compared to the $TEX^H_{86}$-derived temperatures after Kim et al. (2010), which might be due to the relative large number of high latitude core tops on which in this case the BAYSPAR calibration is based, whereas the $TEX^H_{86}$ calibration of Kim et al. (2010) excludes (sub)polar core-

top data. However, both reconstructions show the same trend.

To assess continental organic matter input into the marine realm, the Branched Isoprenoid Tetraether (BIT) index, as proposed by Hopmans et al. (2004), was calculated, including the 6-methyl brGDGTs as described by de Jonge et al. (2014; 2015):

$$BIT = \frac{[brGDGT\ Ia] + [brGDGT\ IIa+IIa'] + [brGDGT\ IIIa+IIIa']}{[brGDGT\ Ia] + [brGDGT\ IIa+IIa'] + [brGDGT\ IIIa+IIIa'] + [Cren]} \qquad (4)$$

where the numbers correspond to different branched GDGTs (Hopmans et al., 2004).

Additionally, we calculated the Methane Index (MI), as a proxy for dissociation of marine gas hydrates conductive to anaerobic oxidation of methane (AOM), as proposed by Zhang et al. (2011):

$$MI = \frac{[GDGT-1] + [GDGT-2] + [GDGT-3]}{[GDGT-1] + [GDGT-2] + [GDGT-3] + [Cren] + [Cren']} \qquad (5)$$

The archaea living in hydrate-impacted environments mainly contain GDGT-1, -2 and -3 as their membrane lipids, and hence relative high abundances of these isomers can indicate these types of environments in the past, and potentially explain erroneous $TEX^{H}_{86}$ results due to high GDGT-1, -2 and/or -3 abundances. For all samples, this index was < 0.3, except for one data point (MI = 0.6), which was therefore removed from further discussion. Additionally, to assess other potential influence on the $TEX^{H}_{86}$, we determined the %GDGT-0 and Ring Index for all sediments. All values were below the advised thresholds, implying no substantial biases on the $TEX_{86}$ (Zhang et al., 2011; Sinninghe Damsté et al., 2012; Zhang et al., 2016).

### 2.5.2 Long chain alkenones

Ketone fractions were dissolved in ethyl acetate to a concentration of ~ 1 mg mL$^{-1}$, and analyzed on an Agilent 6890N gas chromatograph (GC) with flame ionization detection (FID). Separation was achieved on a fused silica column with a length of 50 m and diameter of 0.32 mm, coated with a CP Sil-5 (thickness = 0.12 µm). Helium was used as carrier gas. The flow mode was a constant pressure of 100 kPa. Alkenones were injected at 70 °C at the start of the analysis, increased by 20 °C min$^{-1}$ to 200 °C followed by 3 °C min$^{-1}$ until the final temperature of 320 °C. This end temperature was held for 25 min. Quantification of the alkenones was achieved by means of the $C_{19:0}$ ketone internal standard. Identification of the long chain alkenones was done on an Agilent 7890B GC system interfaced with an Agilent 5977A MS. Separation was achieved on a CP Sil-5 column with an identical diameter and film thickness as that of the GC-FID, but a length of 25 m. Helium was the carrier gas, maintaining a constant flow rate of 2 mL min$^{-1}$. The MS operated at 70 eV. For both systems, the injection volume was 1 µL. The long chain alkenones were identified in full scan, scanning between $m/z$ 50 and 850, and comparison with literature (de Leeuw et al., 1980; Volkman et al., 1980; Marlowe et al., 1984).

The $U^{K'}_{37}$ index was calculated according to Prahl and Wakeham (1987):

$$U^{K'}_{37} = \frac{[C_{37:2}]}{[C_{37:2}] + [C_{37:3}]} \qquad (6)$$

The $U^{K'}_{37}$ values were converted to SSTs using the calibration of Müller et al. (1998):

$$SST = \frac{U^{K'}_{37} - 0.044}{0.033} \qquad (7)$$

There is also a Bayesian calibration available for the $U^{K'}_{37}$, called the BAYSPLINE (Tierney and Tingley, 2018), but below ~ 24 °C temperature estimates are similar to that of the calibration of Müller et al. (1998). Since our temperatures are well below 24 °C, we have applied the calibration of Müller et al. (1998).



### 2.5.3 LCDs

The LCDs were silylated prior to analysis. Polar fractions were dissolved in 25 µL of pyridine and 25 µL of N,O-bis(trimethylsilyl)trifluoroacetamide (BSTFA) and heated at 60 °C for 20 min. Prior to injection, 450 µL ethyl acetate was added. GC-MS analysis was carried out on an Agilent 7890B gas chromatograph coupled to an Agilent 5977A mass spectrometer. Samples were injected at 70 °C. The oven temperature was programmed to 130 °C by 20 °C min$^{-1}$, and subsequently to 320 °C by 4 °C min$^{-1}$; this final temperature was held for 25 min. The GC was equipped with an on-column injector and fused silica column (25 m x 0.32 mm) coated with CP Sil-5 (film thickness 0.12 µm). The carrier gas was Helium at a constant flow of 2 mL.min$^{-1}$. The mass spectrometer operated with an ionization energy of 70 eV. The injection volume was 1 µL. Identification of the LCDs was achieved in full scan, scanning between *m/z* 50 to 850, and on basis of their characteristic fragmentation (Versteegh et al., 1997). Quantification of the LCDs and proxy computation was done by analysis in SIM mode of the characteristic fragments (*m/z* 299, 313, 327 and 341; Rampen et al., 2012; *m/z* 187 for $C_{22}$ 7,16-diol internal standard). We applied the following correction factors for the relative contribution of the selected fragment ions during SIM to the total ion counts: 12% for all saturated diols, 5% for all unsaturated diols and 22% for the $C_{22}$ 7,16-diol internal standard.

Past SST was reconstructed by means of the LDI index (Rampen et al., 2012):

$$\text{LDI} = \frac{[C_{30}\ 1,15-diol]}{[C_{28}\ 1,13-diol] + [C_{30}\ 1,13-diol] + [C_{30}\ 1,15-diol]} \tag{8}$$

Subsequently, LDI values were converted to SST values via the following equation:

$$\text{SST} = \frac{\text{LDI} - 0.095}{0.033} \tag{9}$$

Three samples were discarded for the calculation of the LDI due to partial co-elution of the $C_{28}$ 1,13-diol with another unknown compound.

For the reconstruction of past upwelling conditions, the Diol Index as proposed by Willmott et al. (2010) was applied:

$$\text{Diol Index} = \frac{[C_{28}\ 1,14-diol] + [C_{30}\ 1,14-diol]}{[C_{28}\ 1,14-diol] + [C_{30}\ 1,14-diol] + [C_{28}\ 1,13-diol] + [C_{30}\ 1,13-diol]} \tag{10}$$

We chose to apply the index of Willmott et al. (2010) as the Diol Index according to Rampen et al. (2008) was shown to be also affected by variations in SST due to the inclusion of the $C_{30}$ 1,15-diol in the ratio (Rampen et al., 2014b; Zhu et al., 2018), and since on a glacial-interglacial timescales we expect substantial SST differences, we consider the Willmott et al. (2010) ratio more appropriate.

Possible fluvial input of LCDs was assessed by the fractional abundance of the $C_{32}$ 1,15-diol which is potentially riverine derived (de Bar et al., 2016; Lattaud et al., 2017a; 2017b):

$$F\text{C}_{32}\ 1,15\text{-diol} = \frac{[C_{32}\ 1,15-diol]}{[C_{28}\ 1,13-diol] + [C_{30}\ 1,13-diol] + [C_{30}\ 1,15-diol] + [C_{32}\ 1,15-diol]} \tag{11}$$

The NDI index, a proxy for $PO_4^{3-}$ and $NO_3^-$ concentrations, was calculated following Gal et al. (2018):



$$NDI = \frac{[C_{28}\ 1,14-diol]+[C_{28:1}\ 1,14-diol]}{[C_{28}\ 1,14-diol]+[C_{28:1}\ 1,14-diol]+[C_{30}\ 1,14-diol]+[C_{30:1}\ 1,14-diol][C_{28}\ 1,13-diol]+[C_{30}\ 1,13-diol]+[C_{30}\ 1,15-diol]} \quad (12)$$

The NDI was then translated to $[PO_4^{3-}]$ and $[NO_3^-]$ concentrations by using the following equations (Gal et al., 2018):

$$[PO_4^{3-}] = \frac{NDI - 0.015}{0.413} \quad (13)$$

$$[NO_3^-] = \frac{NDI - 0.075}{0.026} \quad (14)$$

Finally, loliolide and dinosterol were identified by GC-MS analysis (simultaneously with the LCDs) of the silylated polar fraction and quantified using the $C_{22}$ 7,16-diol standard in full scan, correcting for the molecular weights of the compounds.

## 3 Results

### 3.1 Bulk parameters and sedimentation rates

The average TOC content varies between 0.4 and 2.6%. The TOC content is significantly higher during the interglacial periods (MIS 1, 3 and 5) compared to glacial periods (MIS 2, 4 and 6; two-tailed $p < 0.001$): the average TOC concentration is 1.4% for the interglacial periods and 0.7% for the glacial intervals (Fig. 3d). Similarly, the average TN levels are 0.2% and 0.1% during the interglacial and glacial intervals, respectively (Fig. 3d). During Termination 2, both the TOC and TN contents increase rapidly (within < 1 kyr) towards interglacial values, and the highest TOC and TN values are observed around 78 ka (2.6 and 0.3%, respectively). The atomic C/N ratio (Fig. 3b) varies from 4.3 to 10.5 and is on average higher during the interglacial periods (7.2) as compared to glacial times (6.2). The organic matter $\delta^{13}C$ record ($\delta^{13}C_{OM}$) also reveals a glacial-interglacial variation (Fig. 3c), corresponding to slightly, but significantly (at the 5% significance level, two-tailed $p < 0.001$) $^{13}$C-enriched values during interglacial times ($\delta^{13}C_{average} = -21.2‰$) as compared to the glacials ($\delta^{13}C_{average} = -21.8‰$). However, the average difference in $\delta^{13}C$ between glacial and interglacial values over the entire record is relatively small (ca. 0.6‰).

Using the age model modified from Heusser et al. (2006), we estimated sedimentation rates that varied between ca. 0.2 and 1.7 mm yr$^{-1}$ (Fig. 3a). Sedimentation rates were highest in MIS 4, during which the sedimentation rate reaches values of around 2 mm yr$^{-1}$. Sedimentation rates were lowest during the warmest periods (MIS 1 and 5e; ca. 0.3-0.4 mm yr$^{-1}$). The MAR$_{TOC}$ varied between ca. 2 and 32 g m$^{-2}$ yr$^{-1}$ and shows a relatively similar pattern as the sedimentation rate, implying that the sedimentation rate strongly controls the AR of TOC (Fig. 3a). Around ca. 100 ka there is a pronounced maximum in the TOC AR, reaching ca. 32 g m$^{-2}$ yr$^{-1}$.

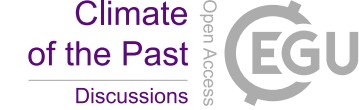

## 3.2 Lipid biomarker concentrations and ARs

In all sediments long chain alkenones, GDGTs and LCDs were present above the quantification limit. The total concentration of crenarchaeol, produced by marine Thaumarchaeota (Schouten et al., 2002; 2003), varies between ca. 2 and 16 µg g$^{-1}$, with highest values during MIS 5, after MIS 5e (Eemian Interglacial; Fig. 4f). ARs range from 1 to 6 mg m$^{-2}$ yr$^{-1}$ (Fig. 4f).

The summed concentration of the di- and tri-unsaturated C$_{37}$ alkenones, a productivity proxy for alkenone-producing haptophytes (e.g., Prahl et al., 1988; Prahl and Muehlhausen, 1989; Rostek et al., 1997; Villanueva et al., 1997), varies between ca. 0.5 and 10 µg g$^{-1}$. Highest concentrations are observed for MIS 5e (126–116 ka; Fig. 4g). Around the boundary of MIS 4 and 5, concentrations decrease from ca. 6 to 2 µg/g sediment. During the Late Holocene, C$_{37}$ alkenone concentrations increase again. The alkenone AR record resembles that of the MAR$_{TOC}$ with a peak around 100 ka of ca. 10 mg m$^{-2}$ yr$^{-1}$ (Fig. 4j).

The main LCDs detected throughout the core are the C$_{28}$ and C$_{30}$ 1,13-, C$_{28}$, C$_{30}$ and C$_{30:1}$ 1,14- and the C$_{30}$ and C$_{32}$ 1,15-diols. We did not detect the C$_{32}$ 1,14-diol (characteristic for *Apedinella radians*), but we did identify the C$_{30:1}$ 1,14 diol and C$_{29}$ 12-OH fatty acid, which are typical biomarkers for *Proboscia* diatoms (Sinninghe Damsté et al., 2003), implying that *Proboscia* diatoms are most likely the source of the 1,14-diols detected in the sediments. The profile of the summed concentration of 1,14-diols shows four distinct sharp peaks during the MIS 5 (Fig 4j), reaching concentrations up to 1.4 µg g$^{-1}$. The 1,14-diol AR shows one distinct peak at 94 ka of ca. 2 mg m$^{-2}$ yr$^{-1}$, whereas throughout the rest of the record the AR varies between 0 and 0.8 mg m$^{-2}$ yr$^{-1}$. Figure 4h shows the ARs of the summed 1,13- and 1,15-diols. The AR of the 1,13-diols ranges between ca. 0.1 and 1.2 mg m$^{-2}$ yr$^{-1}$ throughout MIS 6 and 5, and peaks (ca. 1.5 mg m$^{-2}$ yr$^{-1}$) at the end of MIS 5 revealing highest ARs during MIS 4, followed by a subsequent gradual decrease towards Holocene values of around 0.2 mg m$^{-2}$ yr$^{-1}$. The AR record of the 1,15-diols is highly similar to that of the 1,13-diols for MIS 1 to 4, as well as for MIS 6, but shows higher values during the second half of MIS 5 with maxima around 94 and 104 of between ca. 1 and 1.5 mg m$^{-2}$ yr$^{-1}$. Additionally, we quantified dinosterol, a biomarker for dinoflagellates (Boon et al., 1979; Volkman et al., 1998), as well as loliolide, an indicator of diatom abundance (Klok et al., 1984; Repeta, 1989). Dinosterol shows highest abundance during the Last Interglacial (MIS 5), peaking just after the Eemian Interglacial (between ca. 116 and 110 ka; Fig. 4e) with concentrations of ca. 4 µg g$^{-1}$. This peak is followed by a gradual decrease towards the LGM with values near 1 µg g$^{-1}$ sediment. The AR record of dinosterol highly resembles the concentration record but peaks somewhat later, i.e., between ca. 105 and 98 ka. The loliolide concentration shows one maximum during the Late Holocene of ca. 2.3 ug g$^{-1}$ sediment (Fig. 4d). Its AR varies between ca. 0 and 1 mg m$^{-2}$ yr$^{-1}$, throughout the record, with one pronounced peak around 100 ka of around 2 mg m$^{-2}$ yr$^{-1}$.



### 3.3 Foraminiferal stable isotopic composition

We generated a $\delta^{18}O$ record of *G. bulloides* for MIS 5, from ca. 70 to 142 ka, as well as for a short interval within MIS 3, from ca. 36 to 41 ka, plotted in Fig. 5b. The $\delta^{18}O$ of *G. bulloides* varied between ca. 0.3 and 4.2‰ and are most depleted during the Eemian Interglacial around 123 ka ($\delta^{18}O$ = 0.3‰), and enriched values are observed for MIS 6 and 3.

### 3.4 Organic proxy records

The Diol Index record shows several maxima, of which the most evident one is during MIS 5 (Fig. 4a). During MIS 5e, the Diol Index reveals two peaks around 126 and 116 ka (up to 0.7), followed by a sharp drop and a subsequent increase around 110 ka, reaching a maximum value at ~ 86 ka (up to 0.8). After 86 ka, the index gradually decreases. During MIS 2–4 the Diol

Index varies between ca. 0.1 and 0.5. During MIS 1, the Diol Index shows a gradual increase from ca. 0.15 to 0.35.

We did not detect the mono-unsaturated $C_{28}$ 1,14-diol ($C_{28:1}$ 1,14-diol), and therefore this diol could not be included in the calculation of the NDI index. In ~64% of the surface sediments of the datasets of Rampen et al. (2014b) and de Bar et al. (2016) on which the NDI calibration of Gal et al. (2018) is based, the $C_{28:1}$ 1,14 was also not detected. The NDI record reveals similar variations as the Diol Index, although more pronounced. During MIS 6 the NDI is close to zero, followed by a small

peak at the end of the Eemian Interglacial (NDI = 0.4), coincident with the peak in the Diol Index at 116 ka. NDI-derived $[NO_3^-]$ and $[PO_4^{3-}]$ concentrations around 116 ka are 11 and 0.8 µmol $L^{-1}$, respectively (Fig. 4b). Similar to the Diol index, the ratio then decreases after which the NDI shows a broad peak (~ 100–80 ka) with a maximum at 88 ka (NDI = 0.6), which gives concentrations of 20 and 1.4 µmol $L^{-1}$ when translated to $[NO_3^-]$ and $[PO_4^{3-}]$, respectively (Fig. 4b). Between 80 and 0 ka (i.e., MIS 4–1), the NDI index does not reveal any distinct maxima, and the ratio varies between ca. 0 and 0.2.

Overall the organic SST proxy records broadly follow the same trend, clearly revealing glacial-interglacial temperature variability (Fig. 5b). Indeed, when we cross-correlate the three proxies, all combinations display significant positive correlations ($p < 0.001$; Fig. 6) albeit with some scatter.

Additionally, the SST trends show good correspondence with the planktonic $\delta^{18}O$ record of *G. bulloides* (Fig. 5c), as well as with the benthic oxygen isotope record for ODP Site 1234 (Heusser et al., 2006; Fig 5a), and the global stack of benthic $\delta^{18}O$

values (e.g., Lisiecki and Raymo, 2005; Fig. 5d). Termination 2 (around 130 ka) is clearly expressed in all three organic proxy SST records, showing a rise in temperature of approximately 4 °C within ca. 1 to 2 kyrs. During MIS 5e, SSTs were between ca. 16 and 18 °C. Then, during MIS 5, the interglacial substages representing alternating cold and warm periods are clearly reflected in all three proxy records. The three records show temperature drops between ca. 6 and 7 °C during MIS 5a (ca. 70 ka). During MIS 4, 3 and 2, all SST records show a gradual decrease towards the LGM, during which $U^{K'}_{37}$ and $TEX^H_{86}$ reveal

temperatures between 9 to 10 °C, and the LDI between 7 and 8 °C. Around 25 ka, $U^{K'}_{37}$ and $TEX^H_{86}$ temperatures show a similar steady rise to Holocene temperatures of around 16 and 18 °C. The LDI reveals an approximate 7 °C warming over





Termination 1, starting slightly later at around 22 ka. During the last ~20 kyrs, LDI-derived SSTs are ~3–4 °C lower as compared to the $U^{K'}_{37}$ and $TEX^{H}_{86}$-derived SSTs.

## 4 Discussion

### 4.1 Productivity

The average bulk organic carbon $\delta^{13}C$ over the last 150 kyrs is –21.4 (±0.6; s.dev.)‰, and the atomic C/N ratio displays an average value of 6.8 (±1.3), both implying a dominant marine organic carbon source (Bordovskiy, 1965; Emerson and Hedges, 1988; Meyers, 1997). Today, the amount of sedimentary organic carbon is a good indicator of export production along the Chilean margin (Hebbeln et al., 2000). Hence, higher TOC levels during the interglacial intervals in the ODP 1234 record may suggest enhanced marine productivity during these intervals for this site. However, enhanced interglacial productivity is in

contrast with previous studies that state that coastal productivity on the Peru Margin was highest during the LGM and diminished during the Holocene (e.g., Thomas et al., 1994; Marchant et al., 1999; Thomas, 1999; Hebbeln et al., 2000, 2002; Lamy et al., 2002, 2004; Romero & Hebbeln, 2003; Mohtadi & Hebbeln, 2004). These studies suggest that during the LGM the ACC migrated northward, supplying nutrients (particularly nitrate and phosphate; Levitus et al., 1994), together with a northward shift of the Southern Westerly belt as main precipitation source onshore, resulting in enhanced micronutrient supply

via continental runoff. This combined effect would have stimulated productivity along the Chilean coast during the LGM. Upon deglaciation the climate zones propagated southward resulting in a lowering of productivity, as indicated by pollen, sedimentological and continental studies (e.g., Heusser, 1990; Lamy et al., 1998, 1999, 2001, 2004; Brathauer & Abelmann, 1999; Haberle & Bennett, 2004; Stuut & Lamy, 2004; Kaiser et al., 2005; Heusser et al., 2006) which suggested a 5° to 6° northward movement of the Southern Westerly belt during the LGM. Two studies for core sites at the same latitude as our core

(35–36°S) suggest low paleoproductivity during the LGM (Romero et al., 2006; Mohtadi et al., 2008). Mohtadi et al. (2008) hypothesized that if the climate zones shifted 5 to 6° northward during the LGM, the SWW would be just above the core site (35–36°S), blowing directly onshore thereby preventing coastal upwelling. In turn, during the Early Holocene the subtropical high pressure would have become the dominant atmospheric player according to Romero et al. (2006), favouring upwelling. However, Muratli et al. (2010b), who reconstructed paleoproductivity over the last ~30 kyrs for ODP 1234, suggested that the

rise in TOC after the LGM is probably not the result of increased productivity, but of a lower oxygen availability due to decreased Antarctic Intermediate Water ventilation, and thus increased preservation, since both opal concentrations and opal and organic carbon MARs do not increase simultaneously with TOC concentrations. Chase et al. (2014) support this hypothesis based on the Th normalized organic carbon fluxes. Our (unnormalized) $MAR_{TOC}$ support these findings since although TOC levels show a steady rise after the LGM, we do not observe this increase in the accumulation of organic carbon, and the

$MAR_{TOC}$ is higher during the LGM as compared to the Holocene, suggesting higher glacial productivity. However, during MIS 3, 4 and 5 we observe even higher $MAR_{TOC}$ values, reaching a pronounced maximum around 100 ka during MIS 5, i.e., during an interglacial period. Hence, this record suggests that the general productivity over the last ~150 kyrs was highest



around 100 ka. This interpretation is supported by most of the AR records of the individual biomarker lipids peaking around 100 ka. The $C_{37}$ alkenone, crenarchaeol and dinosterol ARs are all at their maximum between ca. 105 and 95 ka (Fig. 4g, 4f and 4e, respectively), simultaneous with maximal $MAR_{TOC}$ (Fig. 4j). The AR of loliolide (indicating diatom abundance) peaks at 100 ka (Fig. 4d), coincident with a maximum in total diatoms and *Chaetoceros* diatom counts (Mix et al., 2003; Fig 4c), and

concurrent with the peak in $MAR_{TOC}$. *Chaetoceros* diatoms are generally associated with upwelling (e.g., Abrantes, 1988; Abrantes and Moita, 1999), and also for the Peru-Chile upwelling system, the diatom genus *Chaetoceros* is thought to dominate the diatom community during upwelling conditions (e.g., Anabalón et al., 2007; Schrader and Sorknes, 1991; Romero et al., 2001; Vargas et al., 2004; Abrantes et al., 2007; González et al., 2007; Sanchez  et al., 2012). Therefore, the high abundance of *Chaetoceros* diatoms at ~100 ka suggests that there was a peak in upwelling intensity around this time, possibly related to

the southern position of the subtropical high pressure system and associated upwelling-inducing winds. This upwelling would have introduced macronutrients from deeper colder waters into the euphotic zone, stimulating several phytoplankton communities, including haptophytes, dinoflagellates and diatoms.

## 4.2 1,14-diols as past upwelling and nutrient indicators

Interestingly, peaks in the Diol Index and 1,14-diol concentrations and accumulation rates occur before and after (around 116 and 88 ka; Fig. 4a and 4j), but not during the time interval with enhanced upwelling (Fig. 3b and 3c). Contreras et al. (2010) also reported high abundances of the $C_{28}$ 1,14-diol around 120 ka near the coast of Peru (11°S). This suggests that *Proboscia* diatoms were more abundant before and after this period of intense upwelling, and is in agreement with observations that in the present-day Chile-Peru upwelling region, *Proboscia alata* is more dominant when upwelling is less intense (Tarazona et

al., 2003; Herrera and Escribano, 2006). Although the recent time scale is quite different from our long-time record, which represents an integrated signal of several hundreds to thousands of years, it suggests that over the time period 120–80 kyr, on average, upwelling became stronger, reaching a maximum at 100 ka before subsequently decreasing. Shortly before and after the period of maximum upwelling, the conditions (averaged over multiple years) were apparently optimal for *Proboscia* diatoms. As previously suggested by Rampen et al. (2014b), this indicates that the Diol index should perhaps be considered as

a specific indicator for *Proboscia* productivity, rather than upwelling strength generally, as the environmental conditions determining *Proboscia* abundance likely differ from region to region.

Recently, the NDI was introduced as a quantitative paleonutrient proxy. As for the Diol Index and the 1,14-diols AR, the NDI is low during the years of most intense upwelling (around 100 ka), suggesting a minimum in annual mean nitrate and phosphate concentrations, which is highly unlikely. The NDI is based on the saturated and mono-unsaturated $C_{28}$ 1,14-diol relative to

other diols. Whereas we observe a strong correlation between the ARs of the mono-unsaturated and saturated $C_{30}$ 1,14-diol ($R^2 = 0.69$), the correlations between the $C_{28}$ 1,14 and $C_{30}$ 1,14-diol ($R^2 = 0.23$), and the $C_{28}$ 1,14 and $C_{30:1}$ 1,14-diol ($R^2 = 0.18$) are weak, indicating different source organisms for the $C_{28}$ and $C_{30}$ 1,14-diols, in agreement with previous studies (Rampen et al., 2014b; de Bar et al., 2016; Gal et al., 2018). In fact, Rampen et al. (2009) found that 98% of the diols produced



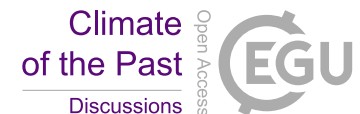

by *P. alata* consists of the saturated and mono-unsaturated $C_{28}$ 1,14-diol, whereas *P. indica* produces similar amounts of the $C_{30}$ and $C_{28}$ 1,14-diol. Furthermore, *P. alata* is the *Proboscia* species detected along the modern coast of Chile and Peru (Tarazona et al., 2003; Herrera and Escribano, 2006), and, therefore, it is likely that *P. alata* is the predominant producer of the $C_{28}$ 1,1,4-diol detected in the ODP 1234 sediments. Consequently, the NDI record likely reflects variations in the abundance

of *P. alata* over the last 150 kyrs, whereas the Diol Index (which also includes the $C_{30}$ 1,14-diol) more likely reflects the abundance of multiple species of *Proboscia*. Previous studies showed that *P. alata* preferentially blooms over other diatoms when nutrients are high but silica concentrations are low (as *P. alata* needs little Si to build its frustule) (Haake et al., 1993; Sakka et al., 1999; Koning et al., 2001; Smith, 2001). In fact, although *P. alata* is often observed in high-nutrient and/or upwelling regions (Hernández-Becerril, 1995; Lange et al., 1998; Koning et al., 2001; Smith, 2001), the conditions

during/under which the species is abundant, is often described as post-bloom, stratification, early upwelling season and/or the oceanic side of the upwelling front (e.g., Hart, 1942; Takahashi et al., 1994; Katsuki et al., 2003; Moita et al., 2003; Tarazona et al., 2003; Herrera and Escribano, 2006; Sukhanova et al., 2006; see references in Table 1 of Rampen et al., 2014b). Moreover, sediment trap studies from the Arabian Sea showed that the maximum flux of *Proboscia* lipids was at the start of the upwelling season (Prahl et al., 2000; Wakeham et al., 2002; Sinninghe Damsté et al., 2003; Rampen et al., 2007). When silicate

concentrations increase (i.e., during upwelling), *P. alata* is likely outcompeted by more heavily silicified diatoms (such as *Chaetoceros*; e.g., Riegman et al., 1996). This suggests that the $C_{28}$ 1,14-diol likely reflects early- or post-upwelling nutrient conditions. Hence, the dip in reconstructed $[PO_4^{3-}]$ and $[NO_3^-]$ concentrations at ~ 100 ka is not realistic and likely due to the low abundance of *P. alata* at this time of intense upwelling. In summary, we suggest that the NDI likely reflects *P. alata* productivity, and may therefore not be suitable as paleo-nutrient tracer.

### 4.3 Sea surface temperature evolution

The three organic proxy-based SST records broadly follow the trend of the planktonic oxygen isotope record for MIS 5, as well as the benthic oxygen isotope record for ODP Site 1234 (Heusser et al., 2006; Fig 5), and the global stack of benthic $\delta^{18}O$ (e.g., Lisiecki and Raymo, 2005), suggesting that the evolution of SST at our study location generally follows global climate

patterns. The resolution of our record is not high enough to recognize possible millennial-scale SST variations related to climatic events such as the Antarctic Cold Reversal. Nevertheless, the overall SST patterns are similar to other Southern Hemisphere records (e.g., Kaiser et al., 2005; Kaiser and Lamy, 2010; Caniupán et al., 2011; Lopes dos Santos et al., 2013) and Antarctic ice-core stable isotope records (e.g., Blunier & Brook, 2001). The three temperature proxies are all significantly positively correlated and present a coherent view of regional climate variability (Fig. 6). In principle, we would expect similar

reconstructed temperatures from the LDI and $U^{K'}_{37}$ indices, since both proxies are based on biomarkers produced by photosynthetic algae (Prahl and Wakeham, 1987; Rampen et al., 2012). Indeed, Kim et al. (2002) showed that for surface sediments off Chile $U^{K'}_{37}$-derived temperatures strongly correlated with annual mean temperatures of the sea surface mixed layer. The linear regression for the $U^{K'}_{37}$ vs. the LDI is close to the 1:1 line, suggesting the LDI is reflecting SST. For the



TEX$_{86}$, it has been shown to potentially reflect subsurface rather than surface water temperatures (Huguet et al., 2007; Kim et al., 2010; 2015; Schouten et al., 2013; Chen et al., 2014) due to the production of isoprenoid GDGTs below the surface mixed layer. Overall, the TEX$^H_{86}$ record agrees reasonably well with the U$^{K'}_{37}$ record for ODP 1234, suggesting that it mainly reflects SST. Also, Kaiser et al. (2015) who established a regional TEX$^H_{86}$ calibration suggested that this proxy mainly reflects SST.

However, all cross-correlations still reveal relatively large scatter, reflecting the multiple different constraints on the respective proxies, as each one is affected by different parameters and can reflect different seasonal temperatures. Nevertheless, for the largest part of the record, the absolute temperature differences between the three temperature proxies is smaller than the maximal possible discrepancy that can be explained by the combined calibration errors (calibration errors of U$^{K'}_{37}$, TEX$^H_{86}$ and LDI are 1.5 °C, 2.5 °C and 2.0 °C, respectively; Müller et al., 1998; Kim et al., 2010; Rampen et al., 2012, respectively).

The only period during which the offsets are larger than calibration errors, is the last ~25 kyrs, i.e., from the LGM to the Holocene, and the relatively brief interval between 52 and 56 ka. During these timespans, the LDI-derived temperatures are ca. 3 to 6 °C lower than those derived from U$^{K'}_{37}$- or TEX$^H_{86}$. This offset could potentially be related to terrestrial input, however, we have assessed the relative contribution of terrestrial derived organic carbon, by means of the BIT index (Hopmans et al., 2004) and the fractional abundance of the C$_{32}$ 1,15-diol (de Bar et al., 2016; Lattaud et al., 2017a; 2017b), which were

both always <0.2, implying that there was no substantial riverine input from terrigenous organic matter. Alternatively, the LDI might be compromised by high input of 1,14-diols (Rodrigo-Gámiz et al. 2015), however, we do not observe a correlation between the LDI (or the SST offset between the LDI and U$^{K'}_{37}$ or TEX$^H_{86}$) and the fractional abundance of the summed 1,14-diols. Interestingly, overall, the LDI reveals greater amplitude over the record as compared to the U$^{K'}_{37}$ and TEX$^H_{86}$, with a maximum temperature difference between the coldest (LGM) and warmest (Eemian Interglacial) temperature of ca. 10.5 °C,

whereas this is ca. 7.5 and 8.0 °C, for the U$^{K'}_{37}$ and TEX$^H_{86}$, respectively. Interestingly, other LDI and U$^{K'}_{37}$ applications on glacial-interglacial timescales also show somewhat greater amplitudes for the LDI records as compared to the U$^{K'}_{37}$, (Rampen et al., 2012; Lopes dos Santos et al., 2013; Rodrigo-Gámiz et al., 2014; Jonas et al., 2017). This might potentially suggest that the surface sediment calibration of the LDI requires some modification, e.g., a lower slope. Further analysis of surface sediments should reveal this.

We can compare our SST records of ODP Site 1234 with those of other SST records along the coast of South America, in particular those generated by the U$^{K'}_{37}$, which has been used the most often (Fig. 7). Our U$^{K'}_{37}$ temperature record agrees well with other records in the vicinity of ODP 1234, both in terms of absolute temperature as well as in glacial-interglacial temperature amplitude. Our U$^{K'}_{37}$ SST record shows an approximate 7 °C warming over Termination 1, which is in agreement with other past SST records for the central and Southern Chilean margin (Kim et al., 2002; Lamy et al., 2002; 2004; Kaiser et

al., 2005; Romero et al., 2006; Kaiser and Lamy, 2010).

Although the timing of deglacial warming associated with Termination 2 is comparable to other records, the last deglaciation (Termination 1) reveals an early start (around 26 ka) at Site 1234 as compared to neighboring sites. The TEX$^H_{86}$ record reveals this same timing, whereas the LDI suggests a timing more comparable to other records in the vicinity (around 20 ka). Regional differences in timing of deglacial warming have previously been related to the southward shift of the SWW, which directly

and indirectly influences local paleoproductivity and upwelling intensity, and thereby potentially leaving site-dependent, unique signatures in the SST records (Mohtadi et al., 2008). Furthermore, the combined regional records show an increase in the magnitude of $U^{K'}_{37}$ temperature variations over glacial-interglacial timescales towards the south (Fig. 7), which agrees with the idea of greater glacial-interglacial temperature differences at high latitudes as compared to low latitudes (e.g., CLIMAP,

1976; Mohtadi & Hebbeln, 2004; Mohtadi et al., 2008).

**5 Conclusions**

We have tested the applicability of long chain diols as tracers for past SST (using the LDI index), upwelling and nutrient conditions (Diol Index), and $[PO_4^{3-}]$ and $[NO_3^-]$ (NDI index) at ODP 1234, located within the Peru-Chile upwelling system. The LDI agrees with the $TEX^H_{86}$ and $U^{K'}_{37}$ sea surface temperature records, as well as with the planktonic $\delta^{18}O$, suggesting

that the LDI reflects past SSTs. During the Last Interglacial (MIS 5), increased accumulation of TOC and phytoplankton lipid biomarkers centered around 100 ka indicate enhanced primary productivity. Concurrently, there is a peak abundance in preserved *Chaetoceros* diatoms, suggesting a peak in upwelling intensity. The Diol index peaks before and after this peak in upwelling, agreeing with present-day diatom distributions along the Chile-Peru margin, with *Chaetoceros* diatoms being dominant during upwelling, and *Proboscia alata* thriving in more stable waters. The NDI (based primarily on the $C_{28}$ 1,14-

diol) shows the same trend as the Diol Index (based on both the $C_{28}$ and $C_{30}$ 1,14-diols), i.e., also showing a dip around 100 ka, suggesting low mean annual nitrate and phosphate concentrations during this upwelling interval, which is not realistic. Likely *P. alata* is the main $C_{28}$ 1,14-diol producer at ODP 1234 suggesting that the NDI likely reflects *P. alata* productivity, and is therefore not suitable as paleo-nutrient tracer, since the species is generally outcompeted when Si concentrations increase, i.e., during upwelling (potentially explaining the minimum in NDI around 100 ka). Overall, these data suggest that

the NDI potentially reflects *Proboscia alata* productivity instead of nutrient concentrations, and that the Diol index should perhaps be considered as a specific indicator for Proboscia productivity, rather than general upwelling conditions.

**Data availability.** The data can be downloaded from PANGAEA.

**Competing interests.** The authors declare that they have no conflict of interest.

**Author contribution.** Marijke W. de Bar, Jaap Sinninghe S. Damsté and Stefan Schouten designed the experiments and Dave Stolwijk carried them out. Jerry F. McManus analysed the planktonic foraminiferal stable oxygen isotopes. Marijke W. de Bar prepared the manuscript with contributions from all co-authors.



**Acknowledgements**

We thank Phil Rumford for sampling of core ODP 1234. Ronald van Bommel is thanked for analytical support. Linda Heusser and Alan Mix are thanked for providing the age model. This research has been funded by the European Research Council (ERC) under the European Union's Seventh Framework Program (FP7/2007-2013) ERC grant agreement [339206] to S.S.
S.S. and J.S.S.D. receive funding from the Netherlands Earth System Science Center (NESSC) though a gravitation grants from the Dutch ministry for Education, Culture and Science (grant number 024.002.001). The contribution of J.F.M. to this work was supported in part by funding from the United States National Science Foundation.

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

**Figure captions**

**Fig. 1**   Maps showing the location of ODP Site 1234. **(a)** A simplified illustration of the present-day atmospheric and oceanographic setting and the site location. SEC = South Equatorial Current; APF = Antarctic Polar Front; STF = Subtropical

Front. **(b)** Present-day mean annual SST (°C) for the region. Data derived from the World Ocean Atlas 2013; maps were drawn in Ocean Data View (Schlitzer, 2015), and modified manually.

**Fig. 2**   Age-depth relations for ODP 1234 derived from Heusser et al. (2006). **(a)** Age-depth constrains based on the correlation of benthic $\delta^{18}$O with Atlantic core MD95204 (dark blue; upper half) and the Vostok ice core chronology (orange;

lower half). Heusser et al. (2006) switched between the two different age models at ~69 ka/~53 mcd (dashed line). However, we used the full upper age model and then linearly interpolated between ~ the age models from 60 to 64 mcd, as indicated by the connecting green line. **(b)** Sedimentation rates calculated after linear interpolation between the age-depth tie-points of the two age models. The sedimentation rates calculated after the linear interpolation between the age models are presented in green. In transparent orange, the sedimentation rates are plotted when applying the dating strategy of Heusser et al. (2006).

**Fig. 3**   Geochemical bulk records for the studied interval of ODP 1234. **(a)** sedimentation rate (blue) and MAR$_{TOC}$ (orange), **(b)** atomic C/N ratio, **(c)** Bulk organic $\delta^{13}$C, **(d)** TOC (red) and TN (green) concentrations. The different color bands indicate different time periods: interglacial stages MIS 1, 3 and 5 in pink, including the MIS 5e in dark pink. The LGM is highlighted in blue. MIS ages are according to Lisiecki and Raymo (2005).

**Fig. 4**   Biomarker proxy and accumulations records for ODP 1234. **(a)** Diol Index (blue) and biogenic opal concentrations (green; Muratli et al., 2010b) for the last ~28 kyrs. **(b)** Phosphate (blue) and nitrate (orange) concentrations calculated after the NDI index (blue). **(c)** Total diatom counts (purple) and *Chaetoceros* diatom counts (blue) (Mix et al., 2003). The thick smoothed lines reflects 2-point running averages of the diatom records. **(d)** Loliolide concentrations and MARs. **(e)** Dinosterol

concentrations and MARs. **(f)** Crenarchaeol concentrations and MARs. **(g)** C$_{37}$ alkenone concentrations and MARs. **(h)** 1,13- and 1,15-diol MARs. **(i)** 1,14-diol concentration and MARs. **(j)** Mass AR of TOC. For panels **(d)** – **(g)** and **(i)**: concentrations are in grey and MARs are in black. The different color bands indicate different time periods: interglacial stages MIS 1, 3 and



5 in pink, including the MIS 5e, in dark pink. The LGM is highlighted in blue. MIS ages are according to Lisiecki and Raymo (2005).

**Fig. 5** Foraminiferal oxygen isotope and organic temperature proxy records for ODP 1234. **(a)** The benthic stable oxygen isotope records for ODP Site 1234 of Heusser et al. (2006). **(b)** The organic geochemical seawater surface temperature records: $U^{K'}_{37}$ (green), $TEX^H_{86}$ (red) and LDI (blue). The thick lines reflect 3-point running averages. **(c)** Planktonic foraminiferal $\delta^{18}O$ of the species *Globigerina bulloides* with the 3-point running average. **(d)** Global compilation of benthic stable oxygen isotopes (Lisiecki and Raymo, 2005). The pink colored bands reflect the interglacial periods (MIS 1, 3 and 5. The pink lines within MIS 5 reflect the different substage $\delta^{18}O$ minima and maxima. MIS (substage) ages are according to Lisiecki and Raymo (2005). The blue band reflects the Last Glacial Maximum, and the star symbol represents the present-day SST (World Ocean Atlas 2013 version 2).

**Fig. 6** Cross-correlations of sea surface temperature estimates based on the $TEX^H_{86}$, $U^{K'}_{37}$ and LDI indices. Linear regressions are indicated by the black solid lines, together with the 95% confidence intervals. All correlations are significant ($p < 0.001$). The orange dashed line represents the 1:1 line.

**Fig. 7** **(a)** Pacific $U^{K'}_{37}$ SST reconstructions off South America for the last ~160 kyrs (3-point running averages). **(b)** Sea surface temperature (°C) map for the region. The colors of the records correspond to the colors of the site symbols in the map. Alkenone-based SST data is from Caniupán et al. (2011) (MD07-3128; 53°S), Ho et al. (2012) (GeoB 3327-5; 43°S), Kaiser et al. (2005) (ODP1233; 41°S), this study (ODP 1234; 36°S), Kim et al., 2002 (GeoB 3302-1 and GIK 17748-2; 33°S), Bin Shaari (2013) (ODP 1237; 16°S), Bin Shaari et al. (2013) (ODP 1239; 1°S) and Bin Shaari et al. (2014) (ODP 1241; 6°N). Surface temperature data derived from the World Ocean Atlas 2013; map is drawn in Ocean Data View (Schlitzer, 2015), and modified manually.



**Figure 1**

**Figure 2**

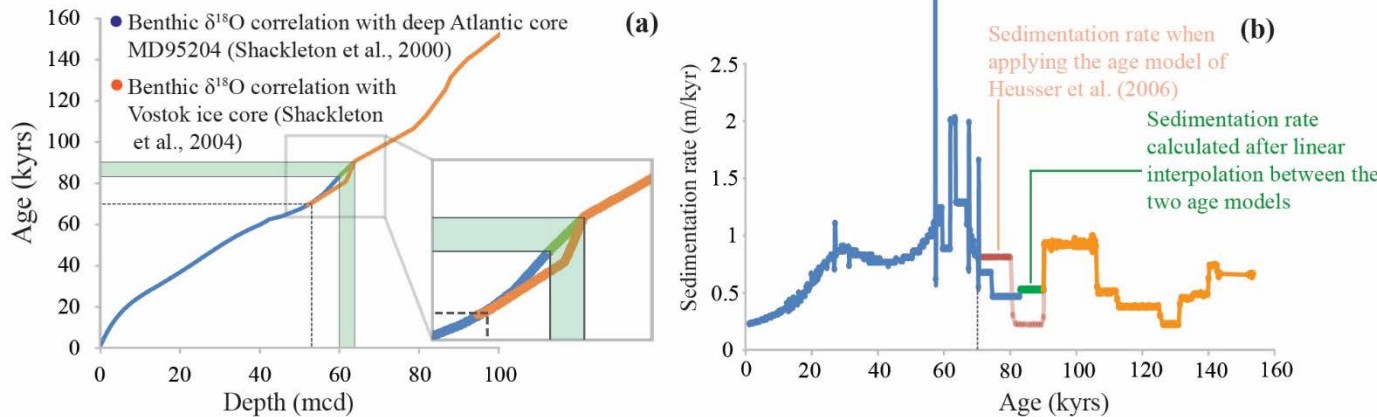





**Figure 3**

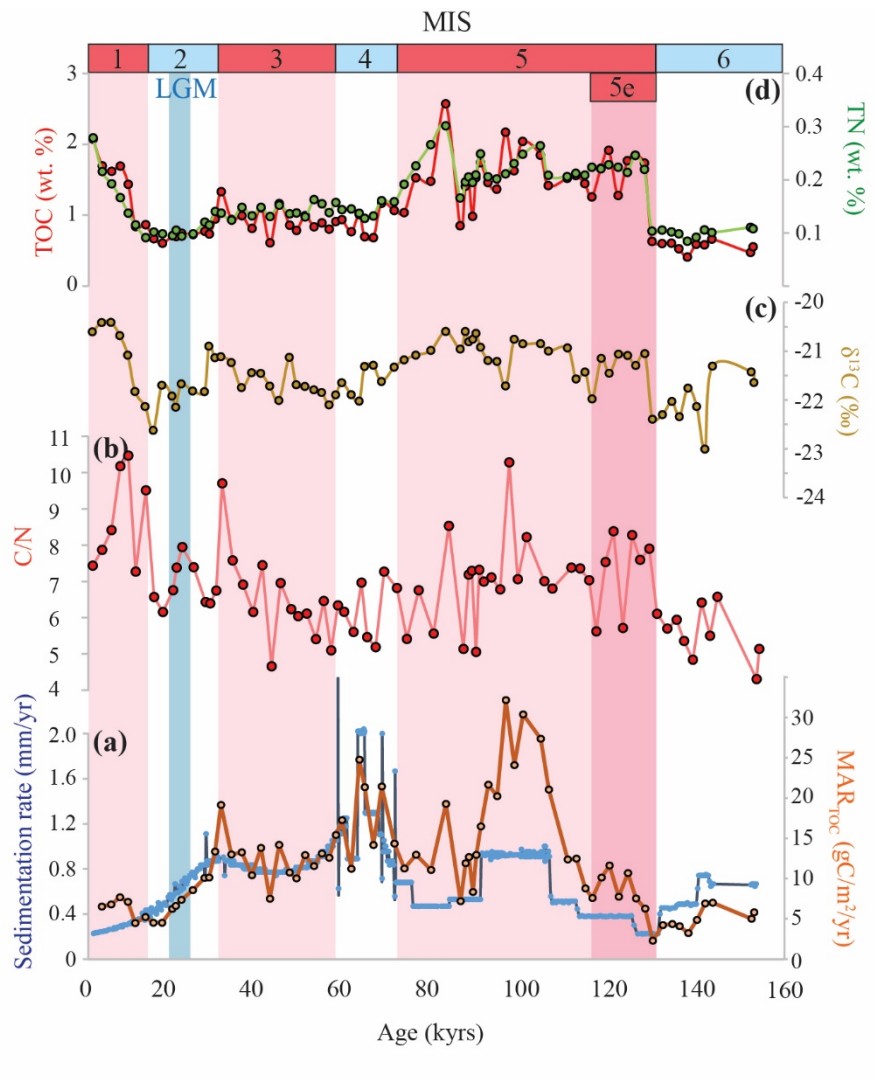





**Figure 4**

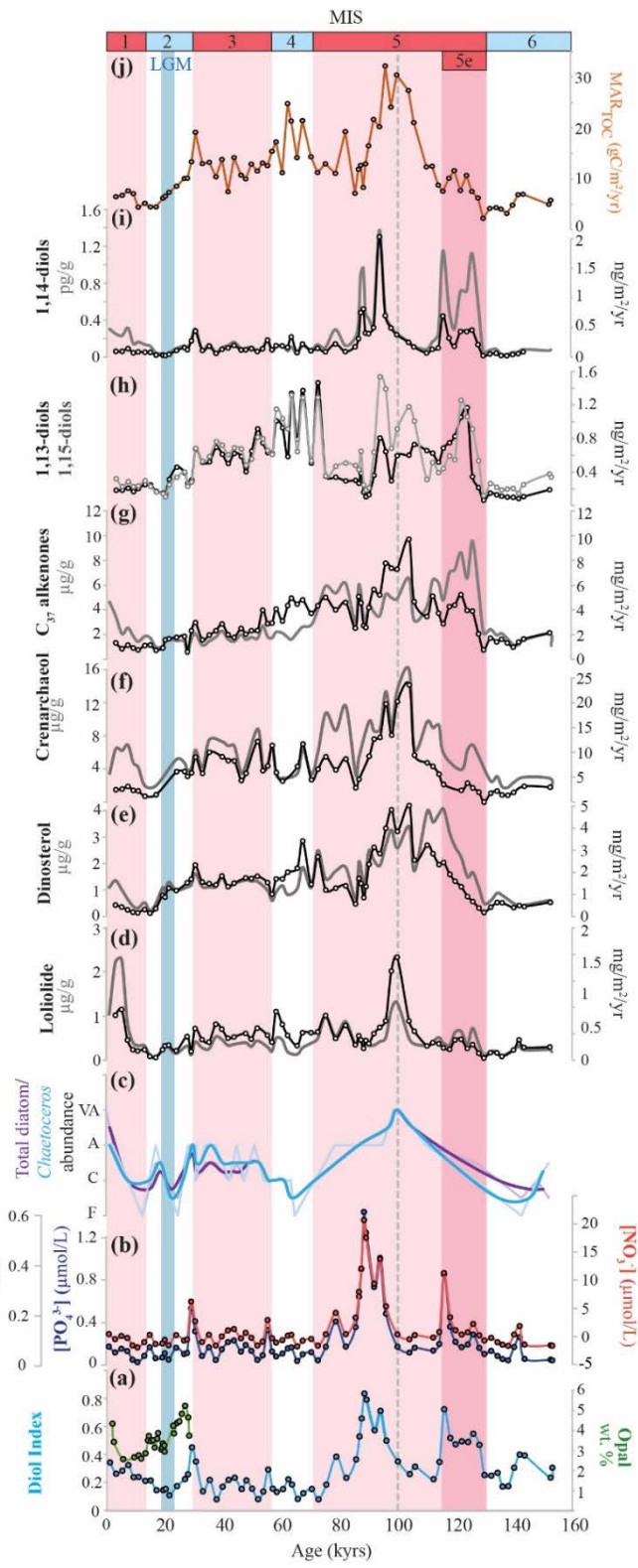



**Figure 5**

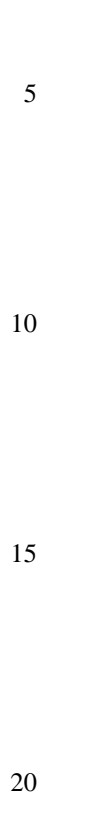
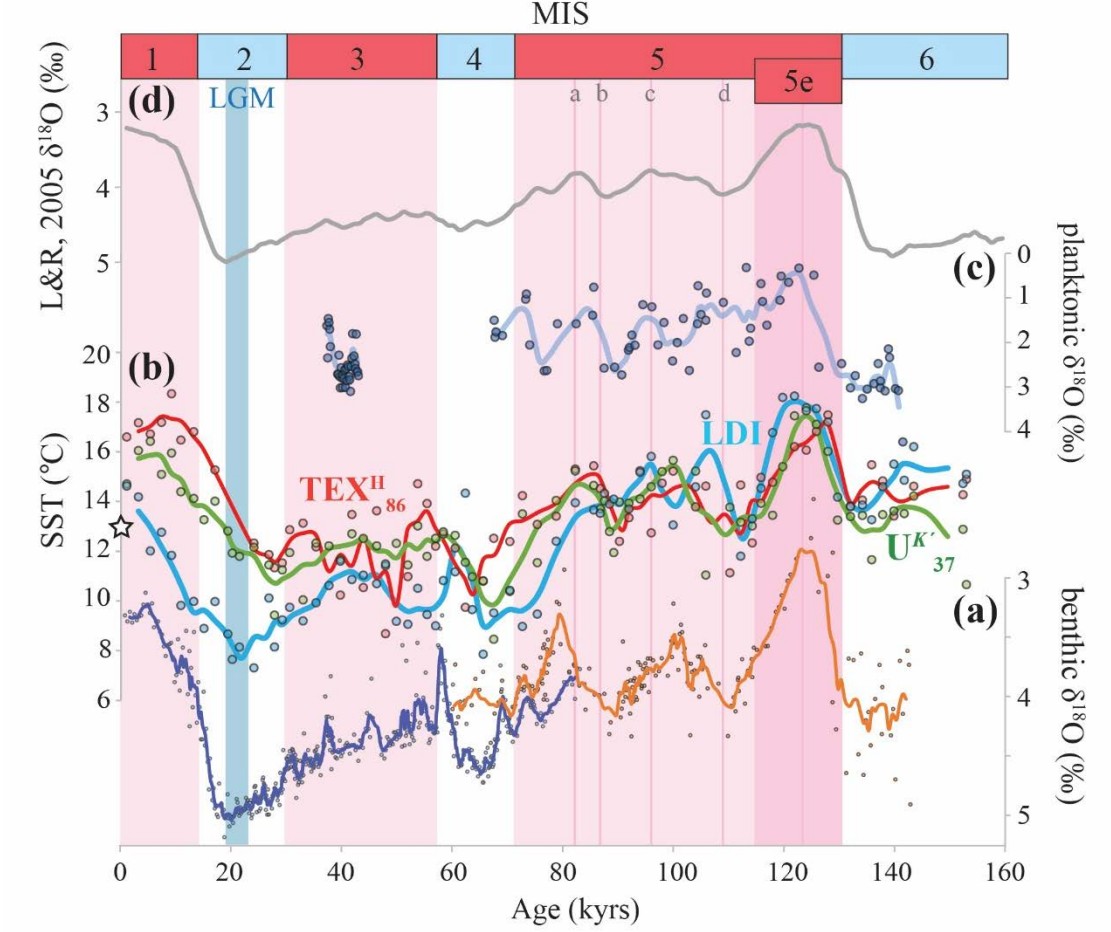





**Figure 6**

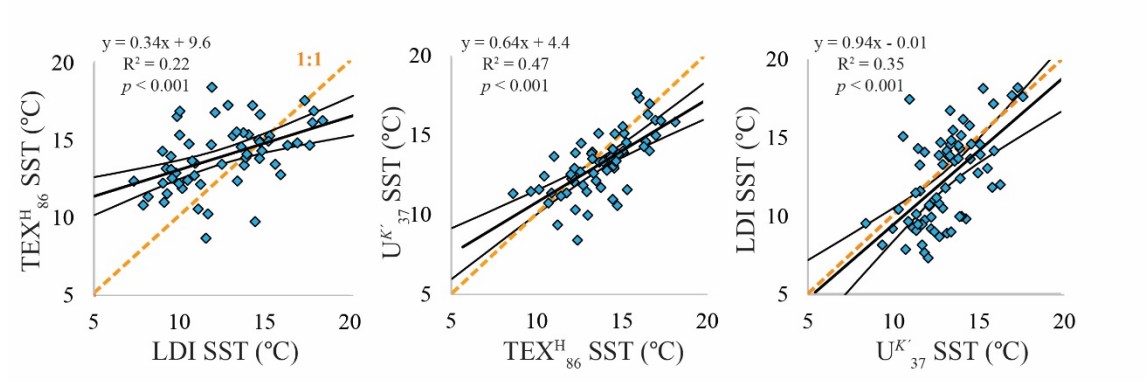

**Figure 7**

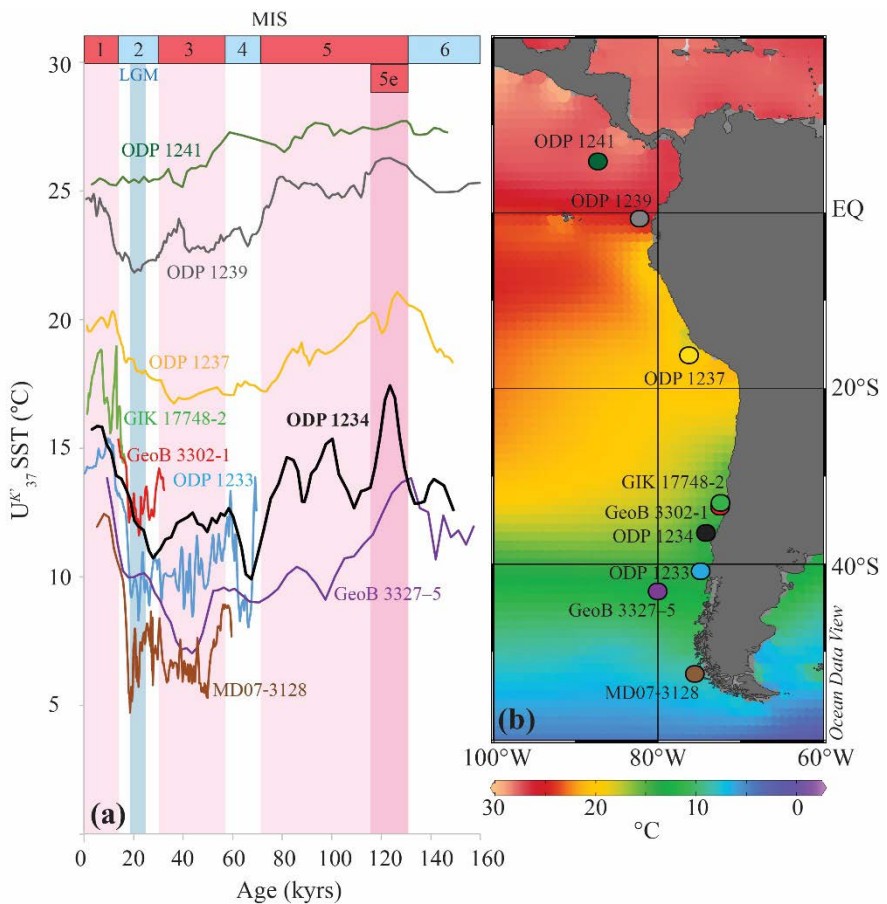



**Appendix A**

**Figure A1.** TEX$_{86}$ records for ODP 1234 applying different calibrations, i.e., the global core-top calibration of Kim et al. (2010) in red (TEX$^H_{86}$), the local core-top calibration of Kaiser et al. (2015) in purple and the Bayesian model of Tierney and Tingley (2014; 2015) in green applying the BAYSPAR tool (http://bayspar.geo.arizona.edu/). For the BAYSPAR calculation we have assumed a prior mean 13 °C and a search tolerance of 0.2 TEX$_{86}$ units. The calculated values are based on TEX$_{86}$ values from 62 20° by 20° latitude-longitude grid boxes.

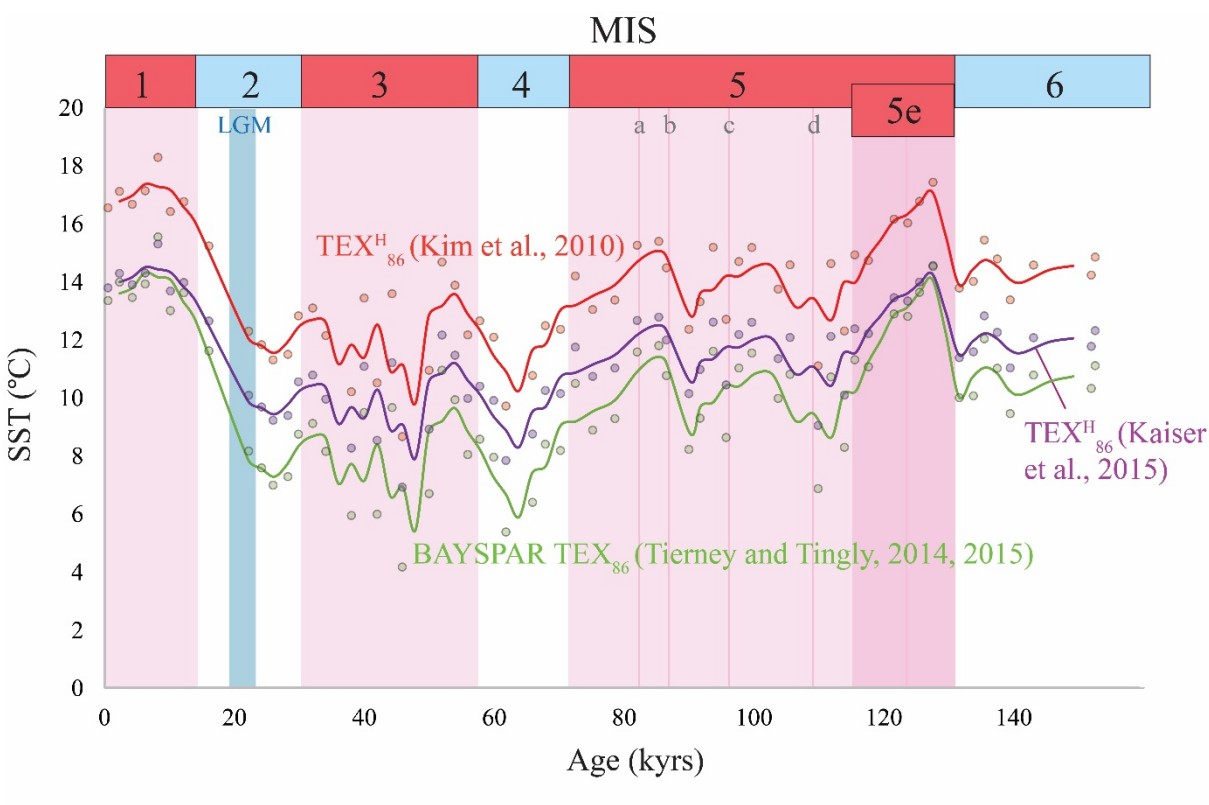