# Peer review of "A Late Quaternary climate record based on long chain diol proxies from the Chilean margin"

_Climate of the Past, 2018_

## Referee Comment (RC1) · Marijke W. de Bar et al. · 12 Sep 2018

1. Does the paper address relevant scientific questions within the scope of CP? YES
2. Does the paper present novel concepts, ideas, tools, or data? Data are novel.
Concepts, ideas and tool have been in several earlier publications. 3. Are substantial
conclusions reached? YES
 4. Are the scientific methods and assumptions valid
and clearly outlined?
The scientific method is clearly outlined. Most of the assump-
tions are properly outlined. 5. Are the results sufficient to support the interpretations
and conclusions? Yes (in general). Some conclusions need to be re-considered (see
detailed review) 6. Is the description of experiments and calculations sufficiently com-
plete and precise to allow their reproduction by fellow scientists (traceability of results)?
YES
 7. Do the authors give proper credit to related work and clearly indicate their

own new/original contribution?  YES 8. Does the title clearly reflect the contents of the paper? YES 9. Does the abstract provide a concise and complete summary?  YES 10. Is the overall presentation well structured and clear?  Mostly 11. Is the language fluent and precise?  It could win from a revision by a native speaker before resubmission. 12. Are mathematical formulae, symbols, abbreviations, and units correctly defined and used?  YES 13. Should any parts of the paper (text, formulae, figures, tables) be clarified, reduced, combined, or eliminated?  The Discussion needs clarification 14. Are the number and quality of references appropriate?  YES 15. Is the amount and quality of supplementary material appropriate?

Comments:

de Bar and co-authors use biomarkers (long chain diols, TEX86 and UK ÌĄ37) to reconstruct sea temperature variations in the river-influenced upwelling ecosystem off southern Chile during the past 150,000 years. They also compare the Diol Index and the nutrient dial index with other paleoproductivity indicators, including bulk organic matter total organic carbon (TOC), organic matter stable carbon isotopes ($\delta$13C), as well as phytoplanktonic lipid biomarkers. The data set is interesting for a broad audience and the technical aspects of the manuscript are correct. In general, the manuscript is well organized. However, (i) there are a several assumptions, which need strong re-thinking and (ii) the MS would have greatly benefited from a quick read by native speaker before submission (several sentences are convoluted and difficult to understand, more a grammar than a scientific problem.) Below I list major comments.

Abstract: all abbreviations should be first fully written. Readers less familiar with them have no clue what the authors are referring to.

Introduction:

l. 15-20: The statement "Proboscia diatoms grow in the early stages of upwelling when nutrients strongly increase in concentration (Koning et al., 2001)" is wrongly interpreted and does not support the authors' interpretation that Proboscia is a diatom indicative of

high productivity in the Chilean coastal upwelling system. If you keep reading Koning et al. (2001), these authors also mention that "The dominance of these pre-upwellers before the onset of the upwelling season was probably caused by their ability to adjust their buoyancy, which allows them to migrate to deeper levels below the euphotic zone to obtain the nutrients trapped there before the actual upwelling starts (Villareal, 1988)." Moreover, "The upwelling period was characterized by the successive dominance of three diatom species, Th. nitzschioides, N. bicapitata and Chaetoceros resting spores. T. nitzschioides dominated the assemblage in July, when the two-gyre upwelling system was firmly established, temperatures were the lowest and H4SiO4 concentrations in the surface waters were high".

Specimens of Proboscia spp. are hardly found in sed traps samples (Romero et al., 2001, Deep-Sea Res. 48, 2673), and in surface and/or downcore sediments along the Chilean margin, and have never been associated with high productivity along the Chilean margin (Romero and Hebbeln, 2003, Mar. Micropal. 48, 71; Mohtadi et al., 2004, J. Quater. Sci., 19, 347; Romero et al., 2006, Quat Res. 65; Mohtadi et al., 2007, Quaternary Sci. Rev. 26, 1055).

P. 3, l. 13: "...several glacial and interglacials periods". Several can be four, but can also be 15. Your study extends only the past 150 kyr, be more concrete.

Results

P. 9, l. 10-15: (i) "The average TOC content varies between 0.4 and 2.6%.": average is not the same as range!; (ii) "The TOC content is significantly higher during the interglacial periods (MIS 1, 3 and 5) compared to glacial periods (MIS 2, 4 and 6)": not quite true, values for MIS 3 are hardly distinguishable from MIS2 and 4. (iii) "During Termination 2...": Terminations should be accordingly identified in Figs 3-5; (iv) "the TOC and TN contents increase rapidly (within < 1 kyr) towards interglacial values": is the sampling resolution high enough to state that the increyse occurred within less than 1,000 years?

P. 9, l. 16-20: very convoluted sentence. Revise.

P. 10, l. 11-25: much of this information is related to Methods. It should be placed accordingly.

P. 11, l. 26: see my comment above for the sampling resolution.

P. 11, l. 26-31: this needs more accurate description. Revise.

Discussion

P. 12, l. 20-32: this part of the Discussion is very intricate and unclear. Please rephrase.

P. 13, l. 5-10: why was no Chaetoceros peak during MIS4 when the MAR TOC was high?

P. 13, l. 17-18 & l. 25-26: since Proboscia is not a secondary component of diatom assemblages in coastal upwelling systems not it is not associated with high productive waters along the Chilean margin, these statements should be thoroughly revised. See my comments above for Introduction

P. 14, l. 2-5: This statement needs appropriate references/lab studies. Have different species of Proboscia been cultured the biomarker content measured in living cells?

P. 14, l. 7: It is not correct stating that "P. alata needs little Si to build its frustule". For diatom standards, frustules of Proboscia are long and build long chains (see Jordan et al., 1991, Diatom Research 6, 63).

P. 14, l. 8-11: These lines contradict your own interpretation of Proboscia as a component of high productive waters diatom assemblages. The fact that P. alata is "often observed in high nutrient and/or upwelling regions" does not mean that this diatom is dominant nor it is a reliable proxy for high productivity.

P. 14, 4.3. Sea surface temperature evolution: the discussion in this section jumps

back and forth between different time windows. This is not reader-friendly. Revise.

P. 14, l. 22-25: Does your SST record following "global climate pattern" refers to MIS5 or the entire record? Please clarify.

P. 15, l. 3: A correlation test helps to supports this statement.

P. 15, l. 7-8: This should be more rigorously discussed.

P. 15, l. 26-27: Looking at your Fig 7, several mismatches in the SST behavior of compared records are recognizable. This should be more critically and rigorously discussed (see for instance ODP1241 and GeoB3327-5 vs ODP1234).

The authors should comments and discussed on: - "The production/export depth of TEXH86 is not well constrained, thus complicating the comparison of TEXH and SST (for example, UK ) based records." (e.g., Kim et al., 2012, EPSL 339, 95-102.; Ho & Laepple, 2018, Nat. Geosc. 9, 606). - "glacial–interglacial amplitude of TEXH86-derived SST change in the tropics is overestimated relative to other proxy evidence, a result also independently found by a multi-proxy study in the subpolar region" (Ho & Laepple, 2015, Earth Planet. Sci. Lett. 409, 15–22; Seki, O. et al. 2014. Prog. Oceanogr. 126, 254–266).

---

## Referee Comment (RC2) · Anonymous Referee #2 · 12 Sep 2018

de Bar at al. test the applicability of different paleoenvironmental proxies based on long chain diols (LDI, Diol Index, and NDI ) by studying the ODP Site 1234 located within the Peru-Chile upwelling system and covering the last 150 kyrs. They compare LDI-derived SSTs with other temperature proxies (TEXH86, UK′37) and with the Diol index and NDI with other phytoplankton production proxies (accumulation rates of TOC and lipid biomarkers). Their results suggest that the Diol Index should not be considered as an upwelling proxy per se, and that the NDI might not be suitable as a more general paleonutrient proxy. I find this is an interesting study. I particularly appreciated the multi-proxy comparison for SST and productivity reconstructions. The overall manuscript is well structured and well written, even though some parts would need

clarification. The data are robust and in general the conclusions are well supported by the data. I however think that some points in the discussion could be clarified/more detailed, as it is sometimes difficult to understand. Please find my comments below.

P. 2, l. 12: "mean annual sea surface temperature" instead of "annual mean sea surface temperature".

P. 4, l. 16: throughout the text you use either "ka" or "kyr". I would be consistent and choose one or another.

P. 9, l. 10: delete "average" as you talk about ranges.

P. 9, l. 13: it would be good to indicate the Terminations on the figures.

P. 10, l. 8-9: the alkenone AR does not show this decrease around the boundary of MIS 4 and 5.

P. 12, l. 16-19: this sentence is not very clear and in contradiction. Please clarify.

P. 12, l. 19-32: this part is not really clear and relatively difficult to follow. Please rephrase.

P. 13, l. 1: "individual lipid biomarkers" instead of "individual biomarker lipids".

P. 13, l. 1-12: How do you explain the peak of MARTOC in MIS 4 which does not correspond to a peak in loliolide concentration/Chaetoceros abundance? There is also no peak for the other lipid biomarkers, except for the 1,13 and 1,15-diols. Who are the potential producers of the 1,13 and 1,15-diols?

P. 15, l. 12-15: this sounds surprising as you state in P. 4, l. 2 that ODP 1234 "lies in the vicinity of two large Andean river systems". Has the influence of terrestrial input on diols been observed for recent times in this site?

P. 15, l. 25-30: you could provide more details on the comparison with the other sites. Even though the general trend agrees well, there are clear differences with sites at

proximity of ODP 1234 (GIK 17748-2; GeoB 3302-1).

P. 28, l. 8: a bracket is missing.

Figures 3, 4, 6, A1: the tick marks on the axes (especially the axis with the Age) are missing. Please add them so the reader can associate more easily the values with the data points.

———————————————

---

## Referee Comment (RC3) · Marijke W. de Bar et al. · 25 Sep 2018

The authors of "A Late Quaternary climate record based on long chain diol proxies from
the Chilean margin" analyzed a suite of organic biomarkers in a marine sediment core
from the coast of Chile to a) assess glacial-interglacial dynamics in marine productiv-
ity (ie. upwelling) and associated climate change, and b) to establish the efficacy of
long-chain diols in paleoenvironmental reconstruction, particularly with regards to the
recently proposed Nutrient Diol Index (NDI) as a metric for marine productivity. They
compared diol-based indices to independent records of SST (alkenones and GDGTs)
and productivity (algal lipid fluxes, total organic carbon fluxes, etc.) and show that NDI
is distinctly different than other upwelling proxies and so is not widely interpretable as
an upwelling indicator, suggesting instead that the principal C28 1,14-diol producer,

[Figure]

Proboscia alata, occurs more commonly in non-upwelling settings. Furthermore, the Diol Index appears to reflect all Proboscia productivity, but still does not capture the upwelling signature expressed in other paleoproductivity proxies. The dataset seems robust and these topics are of broad interest to the paleoceanography and organic geochemistry communities. In general, I think the paper could be published after addressing issues regarding the clarity of the paleoclimate implications and the writing structure.

Please include the full name for each abbreviation the first time abbreviations are mentioned. Likewise, provide the diol-specific equations in the introduction because it would help the reader follow the introduction section easier without flipping through the paper to find the equations.

The goal stated in the first sentence of the abstract stands in contrast with the goal of the title.

Page 12-13: The relationship between temperature and productivity should be discussed more explicitly. I think this section relating the broader climate system and the productivity regime needs clarification. A plot showing the temperature and the nutrient/upwelling indicators together is warranted, given that the paper is drawing links between these two variables. Can either 'northward migration of the SWW' or 'southward migration of the subtropical high' or global cooling induce more upwelling and is there a way to distinguish between these factors in the paleoceanographic record?

Page 14: Many ideas are put forward about P. alata ecology, but I am left with an uncertain idea of what NDI (aka P. alata productivity) variations signify in a specific climatic or ecological sense. Is there a suggestion of why P. alata varies so dramatically during some periods, but not during MIS 1/2/3/4? I think the discussion on the modern observations of P. alata blooms and distributions are helpful, but should be related more explicitly to the paleoceanographic record if possible. Is the author's suggestion that reduced NDI around 100 ka is an indication that upwelling occurred continuously, rather

than on a seasonal basis, thereby eliminating the "early- or post-upwelling nutrient conditions"?

There is an increase in TOC-MAR and importantly, 1,13- and 1,15-diols in MIS 4. Please comment on if this is an indication of specific upwelling or nutrient conditions that might promote the eustigmatophyte but not other algal groups.

P. 15 Starting line 31: Regional differences in the timing of deglacial warming might be expected, but this does not address the different timing among proxies at Site 1234. Please elaborate on the significance of the LDI versus UK37 difference during ∼10 kyr around the LGM.

Some points need grammatical correction:

P1. L. 11: Change "proxies" to "indices" because the Diol Index is stated to be an indicator rather than a proxy in Line 12.

P1. L.12: "...the NDI as a quantitative proxy..."

P1. L. 15: Either specify the number of glacial/interglacial periods or just remove this clause.

P2. L. 16:: Provide the equation for DI, or rephrase, because it is not simply the ratio, but rather the percentage.

P2. L 24: Remove "sea", or change to "marine"?

P2. L 24-25: If possible, specify if this the change in saturation is related to changing species or interspecies diol adjustments.

P2. L. 27: Remove "Preliminary"

P2. L28: "paleo-upwelling" could be used instead of past upwelling. Remove "as". Also, change "proxies" to "indicators" unless the reconstruction of upwelling rates is a quantitative transfer function.

P3. L. 10: The Holocene is part of the Quaternary – no need to mention separately.

P3. L. 10: "large climate cycle" is kind of arbitrary

P3. L. 10: remove "bulk organic matter"

P3. L. 29: Can you include more specifics about the connection – namely, specify if stronger ACC and Westerlies result in warmer or cooler climate.

P. 3 L. 30-33. Does not follow from previous part of paragraph. I think it needs a segue from climate to upwelling/sedimentation dynamics, because this statement isn't about climate like the rest of paragraph. Also, please describe the subtropical high pressure system clearly because it is referred to later as a possible explanation.

P. 4 L. 2: There might be some variation in this, but I think a better spelling is Biobío

P. 4, L. 3: remove "both" and "large" to read ". . .which drain basins of 24,000 km2. . ."

P. 4 L. 13: Remove "providing the potential for high-resolution records" because it's not necessary and I don't think the author's did that kind of sampling.

P. 5 L. 5: If these benthic foraminifera data are already published, I don't think this paragraph is necessary, is it? Are any adjustments made to the McManus and Heusser paper? For brevity, this could be folded into the age model paragraph because it provides the basis for the correlations of the cores. This change is not critical for publishing the paper.

P. 6 L 9: Specify that the GDGT naming scheme in the following descriptions relates to the structures in Castañeda and Schouten, or some other appropriate reference.

P. 6 L. 28: Remove "in this case" and clarify that the BAYSPAR refers to the Tierney and Tingley citation.

P. 6 L. 31-32. Awkward phrasing because of passive voice.

P. 7. "Conductive"?

P. 7, L. 7: "isomers can indicate these types of environments in the past. . ." should be rephrased because it is confusing as written.

P. 7 L. 8: Include a reference suggesting 0.3 is acceptable threshold for Methane Index.

P. 7 L 34: What was the injector configuration?

P. 8 L. 8: Helium does not need to be capitalized.

P. 8 L. 23: The first part of this sentence should be rephrased because it is confusing as written.

P. 8 L. 27: "..potentially riverine derived. . ." should be rephrased for grammar.

P. 8 L. 30: Given that you are testing the effectiveness of the NDI index, insert: "..The NDI index, a proposed proxy for $PO_4^{3-}$. . ."

P. 10 L 34: Insert a comma after "In all sediments"

P. 10: It seems strange that crenarchaeol is the first biomarker presented when it has only been mentioned as a standard in the first part of the paper, and is never discussed or interpreted in any details. Could remove statement about crenarchaeol, else it should be discussed. Similarly, figures 4f and 4g and 4h come before 4a, 4b, etc. in the text. I recommend rearranging this so that diol results come first and things don't seem out of order, or just refer to them as figure 4.

P 10 L 4: The reported accumulation rate for crenarchaeol does not match what is shown in figure 4f.

P. 10 L. 16: Figure "4i", not figure 4j. Also, the data are labeled pg g-1, but the text states $\mu$g g-1. Likewise ng versus mg for the 1,14-diol accumulation rates.

P. 11 L 26-27: State on which proxies these estimates are based.

P. 11, L 28: The spline curves show a decrease of 4-6°C not 6-7°C. Smallest change is for iGDGTs.

P. 12 L 27: "...since neither opal concentration nor opal or TOC-MAR increased simultaneously with TOC concentration" might help clarify this sentence.

P. 13, L 16: Intervals of enhanced upwelling are not explicitly labeled in Figs 3b and 3c.

P. 13, L 27. Rephrase "As for the diol index and the..." because this phrase has another common meaning than as it is used.

P. 14 L 10: "during/under" could just be "in"

Figure 4: How can there be sometimes more chaetoceros counts than total diatom counts?

Figure 5: What are the orange versus blue lines in the Site 1234 Benthic oxygen isotopes? It is not mentioned in the figure caption.

---

## Author Comment (AC3) · 17 Oct 2018

**Response to Referee 3**

The authors of "A Late Quaternary climate record based on long chain diol proxies from the Chilean margin" analyzed a suite of organic biomarkers in a marine sediment core from the coast of Chile to a) assess glacial-interglacial dynamics in marine productivity (ie. upwelling) and associated climate change, and b) to establish the efficacy of long-chain diols in paleoenvironmental reconstruction, particularly with regards to the recently proposed Nutrient Diol Index (NDI) as a metric for marine productivity. They compared diol-based indices to independent records of SST (alkenones and GDGTs) and productivity (algal lipid fluxes, total organic carbon fluxes, etc.) and show that NDI is distinctly different than other upwelling proxies and so is not widely interpretable as an upwelling indicator, suggesting instead that the principal C28 1,14-diol producer, Proboscia alata, occurs more commonly in non-upwelling settings. Furthermore, the Diol Index appears to reflect all Proboscia productivity, but still does not capture the upwelling signature expressed in other paleoproductivity proxies. The dataset seems robust and these topics are of broad interest to the paleoceanography and organic geochemistry communities. In general, I think the paper could be published after addressing issues regarding the clarity of the paleoclimate implications and the writing structure.

**We thank the referee for the positive assessment and for the comments, which we discuss below and will use to improve our manuscript.**

Please include the full name for each abbreviation the first time abbreviations are mentioned. Likewise, provide the diol-specific equations in the introduction because it would help the reader follow the introduction section easier without flipping through the paper to find the equations.

**We will include the full names. However, we do not think that including the diol equations in the introduction will improve the clarity of the text, and we already describe specifically on which LCDs the equations are based.**

The goal stated in the first sentence of the abstract stands in contrast with the goal of the title.

**We agree with the referee, and we will change the title as follows: "Testing the applicability of long chain diol proxies in the Chilean margin for the Late Quaternary"**

Page 12-13: The relationship between temperature and productivity should be discussed more explicitly. I think this section relating the broader climate system and the productivity regime needs clarification. A plot showing the temperature and the nutrient/upwelling indicators together is warranted, given that the paper is drawing links between these two variables. Can either 'northward migration of the SWW' or 'southward migration of the subtropical high' or global cooling induce more upwelling and is there a way to distinguish between these factors in the paleoceanographic record?

**We are not sure on how to more explicitly discuss the relation between productivity and temperature. We start the discussion by illustrating the general idea on how the latitudinal movement of the ACC and SWW linked to the transition of the Last Glacial Maximum to the Holocene might affect productivity. In short, different studies suggest that during the LGM productivity was likely stimulated by greater nutrient input deriving from the ACC (being in the north) and enhanced river runoff due to the northern position of the SWW. Accordingly, productivity would be lower during the Holocene due to the southern position of the ACC and SWW. However, during the Holocene, the winds associated with the subtropical high pressure system might induce upwelling. Moreover, there are also studies that propose that the northern position of the SWW during the LGM might in fact prevent upwelling and thus inhibit productivity despite the greater input of nutrients. Hence, we discuss coastal productivity in light of the main current and wind regimes linked to glacial-interglacial variability. However, most studies have focused on the last ~30 kyrs, and therefore little is still known about upwelling**

and productivity during for instance the Last Interglacial. Nevertheless, we discuss our biomarker records in light of the knowledge that we have of the LGM and Holocene, by for instance proposing that the potential maximum in upwelling intensity around 100 ka, might be related to the subtropical high pressure system which could have stimulated upwelling along the coast, similar as for the Holocene, as proposed by Romero et al. (2006).

Page 14: Many ideas are put forward about P. alata ecology, but I am left with an uncertain idea of what NDI (aka P. alata productivity) variations signify in a specific climatic or ecological sense. Is there a suggestion of why P. alata varies so dramatically during some periods, but not during MIS 1/2/3/4? I think the discussion on the modern observations of P. alata blooms and distributions are helpful, but should be related more explicitly to the paleoceanographic record if possible. Is the author's suggestion that reduced NDI around 100 ka is an indication that upwelling occurred continuously, rather than on a seasonal basis, thereby eliminating the "early- or post-upwelling nutrient conditions"?

**Both the NDI and the Diol Index reflect the relative abundance of *Proboscia* diatoms as the 1,14-diols are specific for this group of diatoms. Since along the Chilean margin *Proboscia alata* has been observed, we propose that for our location the NDI reflects *P. alata* productivity. However, in other regions the $C_{28}$ 1,14-diol might be produced by other *Proboscia* species and the NDI might thus not reflect *P. alata* productivity. In this light we will also partly adjust our hypothesis that the NDI potentially reflects *P. alata* productivity; it is merely more species-specific as compared to the Diol Index since it excludes the $C_{30}$ 1,14-diol from the numerator.**
**As for the question why the NDI/*Proboscia* activity is relatively constant during MIS 1/2/3/4 but shows large variation during MIS 5: this is one of the main issues we have addressed in the manuscript. We propose, based on the maximum in $MAR_{TOC}$ during MIS 5 concurrent with e.g. the highest abundances of preserved *Chaetoceros* valves, that around 100 ka the upwelling intensity was likely to be strong and primary productivity was stimulated. What we know from the modern-day *Proboscia* ecology, is that this diatom genus proliferates when nutrients increase, i.e. during upwelling. However, it is particularly able of dominating the diatom pool during early upwelling conditions when silicate concentrations are still low, and the nutrients are still in the deeper waters. When actual upwelling occurs, and the nutrients are transported to the surface and silicate concentrations increase, *Proboscia* is likely outcompeted by for instance *Chaetoceros* which is more heavily silicified. Accordingly, we think this might explain the relatively low Diol Index and NDI around 100 ka, as the relative abundance of *Proboscia* was likely to be low. However, before and after this event, i.e. around ca. 120 and 90 ka, upwelling was likely less intense, and conditions were more favorable for *Proboscia*.**
**As for the referee's last question: the reduced NDI around 100 ka does not indicate that upwelling occurred continuously throughout the year and thus does not eliminate the seasonal basis of upwelling. However, the NDI and Diol Index reflect integrations of multiple years and thus an averaged signal of *Proboscia* productivity over these years. We believe that the minimum in the NDI around 100 ka, suggesting that phosphate and nitrate concentrations were around 0 µmol/L, is not realistic, because the "mean annual nutrient concentrations" were not likely to be ~0 during this upwelling interval. Even though nutrients can be used up at the end of an upwelling season, mean annual nutrient concentrations are likely to be relatively high.**

There is an increase in TOC-MAR and importantly, 1,13- and 1,15-diols in MIS 4. Please comment on if this is an indication of specific upwelling or nutrient conditions that might promote the eustigmatophyte but not other algal groups.

**Yes, this is true, and we will discuss this in the revised manuscript. Moreover, similar comments were put forward by referees 1 and 2. We believe that the increase in $MAR_{TOC}$ during MIS 4 is likely linked to the evident increase in sedimentation rate during this period (Fig. 3a). During MIS 5, the maximum in $MAR_{TOC}$ corresponds to elevated TOC levels and a constant**

**sedimentation rate, suggesting an increase in primary productivity. However, during MIS 4, the TOC levels are almost twice as low, but the sedimentation rate is clearly higher, suggesting that the increased $MAR_{TOC}$ is a result of enhanced particle settling.**

P. 15 Starting line 31: Regional differences in the timing of deglacial warming might be expected, but this does not address the different timing among proxies at Site 1234. Please elaborate on the significance of the LDI versus UK37 difference during ~10 kyr around the LGM.

**Yes, we completely agree with referee 3, and in our new version of the manuscript we will try to elaborate on this. However, we are not certain on how to explain this ±10 kyr time lag between the deglacial warming as recorded by the LDI and the $U^{K'}_{37}$ and $TEX^{H}_{86}$. The LDI-temperatures are also considerably lower as compared to the other proxy temperatures. Although, the producers of the 1,13- and 1,15-diols are unknown, they are likely to be phototrophs living in the upper part of the photic zone (e.g., Rampen et al., 2012; Balzano et al., 2018), and therefore we would expect similar temperature estimates for the LDI and $U^{K'}_{37}$. Potentially, due to regional climatic change associated with the southward migration of the ACC and SWW upon the deglacial warming, the main season of production of the 1,13- and 1,15-diols shifted, resulting in lower reconstructed SSTs. The difference in timing of the deglacial warming stays elusive.**

Some points need grammatical correction:

P1. L. 11: Change "proxies" to "indices" because the Diol Index is stated to be an indicator rather than a proxy in Line 12.

**We will adjust this.**

P1. L.12: ". . .the NDI as a quantitative proxy. . ."

**We will rephrase as follows: "… *the NDI index as proxy for…*".**

P1. L. 15: Either specify the number of glacial/interglacial periods or just remove this clause.

**We will adjust this.**

P2. L. 16:: Provide the equation for DI, or rephrase, because it is not simply the ratio, but rather the percentage.

**We agree that this formulation is unclear, and we will rephrase this.**

P2. L 24: Remove "sea", or change to "marine"?

**We will change this in "marine".**

P2. L 24-25: If possible, specify if the change in saturation is related to changing species or interspecies diol adjustments.

**In section 4.2 (page 14, around line 1-5) we will add some information on the (un)saturation of the 1,14-diols. In sediments, mono-unsaturated 1,14-diols are generally lower in abundance as compared to the saturated 1,14-diols. However, culture data show that *Proboscia* diatoms often produce more unsaturated diols, and that the degree of unsaturation is strongly affected by temperature (Rampen et al., 2009; OG). This temperature relationship was however not observed in marine surface sediments (Rampen et al., 2014) and the low abundance of unsaturated 1,14-diols might be the result of preferential degradation. However, the regional differences in 1,14-diol distributions are likely also the result of different species producing different 1,14-diols.**

P2. L. 27: Remove "Preliminary"

**We will remove this.**

P2. L28: "paleo-upwelling" could be used instead of past upwelling. Remove "as". Also, change "proxies" to "indicators" unless the reconstruction of upwelling rates is a quantitative transfer function.

**We will correct this.**

P3. L. 10: The Holocene is part of the Quaternary – no need to mention separately. orzet

**We will remove "and Holocene".**

P3. L. 10: "large climate cycle" is kind of arbitrary

**We will remove "large".**

P3. L. 10: remove "bulk organic matter"

**We will remove this.**

P3. L. 29: Can you include more specifics about the connection – namely, specify if stronger ACC and Westerlies result in warmer or cooler climate.

**The relationship between the ACC and Westerlies with productivity and upwelling is described in section 4.1, but we agree that we should elaborate more on the link between the ACC/Westerlies and temperature, which we will do in the new manuscript, something along the lines of:**

**Lamy et al. (2002) showed that millennial- to multicentennial scale variations in past temperature and productivity lags variations in rainfall, and thus the latitudinal migration of the Westerlies. However, the authors found a strong relationship between past seawater temperature variability and Antarctic climate change, and thus the latitudinal position of the ACC. When the ACC migrates northward, potentially associated with an expansion of Antarctic sea ice, cold, subantarctic waters are advected into the southern hemisphere midlatitudes, including the Chilean margin (e.g., Lamy et al., 2004; Kaiser et al., 2005).**

P. 3 L. 30-33. Does not follow from previous part of paragraph. I think it needs a segue from climate to upwelling/sedimentation dynamics, because this statement isn't about climate like the rest of paragraph. Also, please describe the subtropical high pressure system clearly because it is referred to later as a possible explanation.

**We agree that this structure is not optimal, and we will reorganize. Additionally, we will describe the subtropical high pressure system as well, as this is indeed required for the discussion.**

P. 4 L. 2: There might be some variation in this, but I think a better spelling is Biobío

**We will correct this.**

P. 4, L. 3: remove "both" and "large" to read ". . .which drain basins of 24,000 km2. . ."

**We will adjust this.**

P. 4 L. 13: Remove "providing the potential for high-resolution records" because it's not necessary and I don't think the author's did that kind of sampling.

**We will remove this.**

P. 5 L. 5: If these benthic foraminifera data are already published, I don't think this paragraph is necessary, is it? Are any adjustments made to the McManus and Heusser paper? For brevity, this could be folded into the age model paragraph because it provides the basis for the correlations of the cores. This change is not critical for publishing the paper.

**The correction of 0.64‰ is new information and therefore relevant. We agree that this might be better suited in the age model section.**

P. 6 L 9: Specify that the GDGT naming scheme in the following descriptions relates to the structures in Castañeda and Schouten, or some other appropriate reference

**We will clarify this.**

P. 6 L. 28: Remove "in this case" and clarify that the BAYSPAR refers to the Tierney and Tingley citation.

**We believe that "in this case" is relevant here, since the core-tops on which the calibration is based, is different for every paleo-reconstruction; it depends on the assumed prior mean temperature and search tolerance. We used a prior mean of 13 °C and a search tolerance of 0.2 TEX$_{86}$ units, resulting in the 62 high latitudinal grid boxes on which in this case the calibration is based. We will refer to Tierney and Tingley (2014, 2015).**

P. 6 L. 31-32. Awkward phrasing because of passive voice.

**We will rephrase this sentence.**

P. 7. "Conductive"?

**We thank the referee for remarking this typo: it should be "conducive".**

P. 7, L. 7: "isomers can indicate these types of environments in the past. . ." should be rephrased because it is confusing as written.

**We will rephrase this sentence.**

P. 7 L. 8: Include a reference suggesting 0.3 is acceptable threshold for Methane Index.

**We refer here to Zhang et al. (2011): "*Combined GDGT distribution and isotopic evidence suggest that 0.3 to 0.5 of the MI might be a reasonable threshold for distinguishing gas hydrate impacted and/or methane-rich environments from normal marine realm.*"**

P. 7 L 34: What was the injector configuration?
**On-column injection, we will clarify this.**

P. 8 L. 8: Helium does not need to be capitalized.
**We will correct this.**

P. 8 L. 23: The first part of this sentence should be rephrased because it is confusing as written.
We will rephrase this part.

P. 8 L. 27: "..potentially riverine derived. . ." should be rephrased for grammar.
**We will change as follows: "*which is potentially derived from rivers*".**

P. 8 L. 30: Given that you are testing the effectiveness of the NDI index, insert: "..The NDI index, a proposed proxy for PO43-. . ."
**We will adjust this.**

P. 10 L 34: Insert a comma after "In all sediments"
**We will correct this.**

P. 10: It seems strange that crenarchaeol is the first biomarker presented when it has only been mentioned as a standard in the first part of the paper, and is never discussed or interpreted in any details. Could remove statement about crenarchaeol, else it should be discussed. Similarly, figures 4f and 4g and 4h come before 4a, 4b, etc. in the text. I recommend rearranging this so that diol results come first and things don't seem out of order, or just refer to them as figure 4.

**We will clarify that crenarchaeol is a specific biomarker for Thaumarchaeota and thus is indicative for Thaumarcheotal productivity. We agree that the order in Figure 4 is in contrast with the results, and we will rearrange the panels in this figure.**

P 10 L 4: The reported accumulation rate for crenarchaeol does not match what is shown in figure 4f.

**We thank the referee for noticing this, we will correct this.**

P. 10 L. 16: Figure "4i", not figure 4j. Also, the data are labeled pg g-1, but the text states µg g-1. Likewise ng versus mg for the 1,14-diol accumulation rates.
**We thank the referee for pointing out these inaccuracies, and we will correct these.**

P. 11 L 26-27: State on which proxies these estimates are based.
**We agree that this was unclear, and we will clarify this.**

P. 11, L 28: The spline curves show a decrease of 4-6∘C not 6-7∘C. Smallest change is for iGDGTs.

**We will correct this.**

P. 12 L 27: ". . .since neither opal concentration nor opal or TOC-MAR increased simultaneously with TOC concentration" might help clarify this sentence.

**We will change accordingly.**

P. 13, L 16: Intervals of enhanced upwelling are not explicitly labeled in Figs 3b and 3c.

**We agree that this is unclear and we will change as follows: "… *with enhanced upwelling (i.e., around 100 ka; dashed line in Fig. 3)*".**

P. 13, L 27. Rephrase "As for the diol index and the. . ." because this phrase has another common meaning than as it is used.

**We will rephrase this sentence.**

P. 14 L 10: "during/under" could just be "in"

**We will correct this.**

Figure 4: How can there be sometimes more chaetoceros counts than total diatom counts?

**This is because the counts are not quantitative, but descriptive: F=few; C=common; A=abundant; VA=very abundant. However, we now realize that we did not clarify this is in the caption, which we will correct in the new version.**

Figure 5: What are the orange versus blue lines in the Site 1234 Benthic oxygen isotopes? It is not mentioned in the figure caption.

We thank the referee for noticing this, and we will clarify this in the caption. The color scheme is similar to Fig. 2, where the two colors reflect the two separate age models. The blue data reflect the benthic $\delta^{18}$O dated by correlation to Atlantic core MD95204, whereas the orange data represent the $\delta^{18}$O data correlated to the Vostok ice core chronology.

---

## Author Response (AR1)

**Response to reviewers**

**Submission cp-2018-88 to *Climate of the Past**

*Editor:*

Dear Dr. de Bar and co-authors,

Thank you for your substantial efforts in responding to the comments of the three reviewers. The reviewers provided extensive but constructive comments on your manuscript, and there is general consensus that the dataset is interesting and the conclusions are relevant to the paleoceanography and organic geochemistry readership of Climate of the Past.

My sense is that your responses address the main technical, interpretive, and clarity concerns of the three reviewers, without overly broadening the scope the manuscript. Thus, I encourage you to formally submit a revised manuscript that incorporates the changes indicated in your detailed responses to the reviewers.

Sincerely,

Alberto Reyes

**We thank the editor for the positive assessment of our manuscript. We have revised the manuscript and below we provide point-to-point answers to the comments of the reviewers: when applicable, we indicated where adjustments were made in the text (note: when we refer to line numbers in which we have made adjustments, we refer to the line numbering of the revised manuscript with "track changes" in this document). The reviewers' comments are in regular font; our replies are in bold font.**

**Sincerely, also on behalf of all co-authors,**

**Marijke de Bar**

*Editor:*

After assessing the reviewer comments, I have a few small suggestions too:

1. Perhaps some of the criticisms from Rev 1 and 3 about lack of clarity in parts of the discussion in p.12-14 could be partially resolved by breaking some of your longer paragraphs into shorter, more focused paragraphs.

**We agree, and we have now structured the discussion in a clearer way.**

2. Reviewer 1 suggested several topics for expanded discussion, but I'm convinced by your responses that extensive additions run the risk of overly broadening the scope of the manuscript.

3. ka vs kyr: It seems you've used ka = "thousands of years *ago*" and kyr = "thousand years". This is a useful distinction so I suggest sticking with it.

**We have followed these suggestions.**

**Response to Referee 1**

Comments: de Bar and co-authors use biomarkers (long chain diols, TEX$_{86}$ and U$^{K'}_{37}$) to reconstruct sea temperature variations in the river-influenced upwelling ecosystem off southern Chile during the past 150,000 years. They also compare the Diol Index and the nutrient dial index with other paleoproductivity indicators, including bulk organic matter total organic carbon (TOC), organic matter stable carbon isotopes (δ13C), as well as phytoplanktonic lipid biomarkers. The data set is interesting for a broad audience and the technical aspects of the manuscript are correct. In general, the manuscript is well organized. However, (i) there are a several assumptions, which need strong re-thinking and (ii) the MS would have greatly benefited from a quick read by native speaker before submission (several sentences are convoluted and difficult to understand, more a grammar than a scientific problem.) Below I list major comments.

**We thank the referee for the positive assessment and for the comments, which we have seriously considered.**

Abstract:

all abbreviations should be first fully written. Readers less familiar with them have no clue what the authors are referring to.

**We have adjusted this.**

Introduction:

l. 15-20: The statement "Proboscia diatoms grow in the early stages of upwelling when nutrients strongly increase in concentration (Koning et al., 2001)" is wrongly interpreted and does not support the authors' interpretation that Proboscia is a diatom indicative of high productivity in the Chilean coastal upwelling system. If you keep reading Koning et al. (2001), these authors also mention that "The dominance of these pre-upwellers before the onset of the upwelling season was probably caused by their ability to adjust their buoyancy, which allows them to migrate to deeper levels below the euphotic zone to obtain the nutrients trapped there before the actual upwelling starts (Villareal, 1988)." Moreover, "The upwelling period was characterized by the successive dominance of three diatom species, Th. nitzschioides, N. bicapitata and Chaetoceros resting spores. T. nitzschioides dominated the assemblage in July, when the two-gyre upwelling system was firmly established, temperatures were the lowest and H4SiO4 concentrations in the surface waters were high"

**We do not state that *Proboscia* is indicative of high productivity along the Chilean margin. Instead we refer to Tarazona et al. (2003) and Herrera and Escribano (2006) who both describe *Proboscia alata* as being dominant when upwelling is less intense, and thus when general productivity is likely to be lower. Moreover, we suggest that the Diol Index should perhaps be considered more as an indicator of *Proboscia* productivity, rather than general productivity.**

**Concerning the reference to Koning et al. (2001), we have rephrased the sentence as follows (P. 2, lines 19-20): "Proboscia grows during the early stages of upwelling since they need relatively little silica and they are able to migrate to deeper waters to obtain nutrients."**

Specimens of Proboscia spp. are hardly found in sed traps samples (Romero et al., 2001, Deep-Sea Res. 48, 2673), and in surface and/or downcore sediments along the Chilean margin, and have never been associated with high productivity along the Chilean margin (Romero and Hebbeln, 2003, Mar. Micropal. 48, 71; Mohtadi et al., 2004, J. Quater. Sci., 19, 347; Romero et al., 2006, Quat Res. 65; Mohtadi et al., 2007, Quaternary Sci. Rev. 26, 5   1055).

**This is correct. However, the main reason for its rare occurrence in sediments and sediment trap is likely the weak preservation potential (Jordan and Priddle, 1991; Koç et al., 2001; Jordan and Ito, 2002). Furthermore, we also do not claim that *Proboscia* is a dominant diatom genus along the Chilean margin, and as stated above, we do not claim that *Proboscia* is indicative of high productivity in this region.**

P. 3, l. 13: "...several glacial and interglacials periods". Several can be four, but can also be 15. Your study extends only the past 150 kyr, be more concrete.

**We have removed "several".**

Results
P. 9, l. 10-15: (i) "The average TOC content varies between 0.4 and 2.6%.": average is not the same as range!; (ii) "The TOC content is significantly higher during the interglacial periods (MIS 1, 3 and 5) compared to glacial periods (MIS 2, 4 and 6)": not
20   quite true, values for MIS 3 are hardly distinguishable from MIS2 and 4. (iii) "During Termination 2. . .": Terminations should be accordingly identified in Figs 3-5; (iv) "the TOC and TN contents increase rapidly (within < 1 kyr) towards interglacial values": is the sampling resolution high enough to state that the increyse occurred within less than 1,000 years?

**(i) We have corrected this.**
**(ii) We agree that this is not formulated clearly, and we will adjust this. However, the average TOC levels (taken together for MIS 1, 3 and 5) of ca 1.4% are in fact significantly higher as compared to the average level for MIS 2, 4 and 6 which is ca. 0.7%. However, we have now added the following (P. 9, lines 24-27):**
30   **"*The TOC content is significantly higher during the interglacial periods (MIS 1, 3 and 5) compared to glacial periods (MIS 2, 4 and 6; two-tailed p < 0.001): the average TOC concentration is 1.4% for the interglacial periods and 0.7% for the glacial intervals (Fig. 3d), although TOC levels for MIS 3 are quite similar to those of MIS 1 and 2.*"**
**(iii) We have indicated the terminations in Figures 3-5.**
35   **(iv) We thank the reviewer for pointing this out, as this should indeed be "within < 2 kyrs". We have corrected this.**

P. 9, l. 16-20: very convoluted sentence. Revise

40   **We have revised this sentence as follows (P. 10, lines 5-8):**
**"*The organic matter $\delta^{13}C$ record ($\delta^{13}C_{OM}$) also reveals a glacial-interglacial variation (Fig. 3c), corresponding to slightly $^{13}C$-enriched values during interglacial times ($\delta^{13}C_{average} = -21.2‰$) as compared to the glacials ($\delta^{13}C_{average} = -21.8‰$). Although small, these changes are statistically significant (5% significance level, two-tailed p < 0.001).*"**

P. 10, l. 11-25: much of this information is related to Methods. It should be placed accordingly.

**We have moved the following sentence to the Methods section (P. 9 lines 17-18): "*Additionally, we quantified dinosterol, a biomarker for dinoflagellates (Boon et al., 1979; Volkman et al., 1998), as well as loliolide, an indicator of diatom abundance (Klok et al., 1984; Repeta, 1989).*"**

P. 11, l. 26: see my comment above for the sampling resolution.

**We have corrected this.**

P. 11, l. 26-31: this needs more accurate description. Revise.
**Since we merely want to compare our LDI record with the $U^{K'}_{37}$, $TEX^{H}_{86}$ and $\delta^{18}O$ records to assess whether the LDI is suitable as SST proxy in this region, we believe that our result description is sufficient.**

Discussion

P. 12, l. 20-32: this part of the Discussion is very intricate and unclear. Please rephrase.
**We agree that this part in difficult to read, and we have tried to improve the readability.**

P. 13, l. 5-10: why was no Chaetoceros peak during MIS4 when the MAR TOC was high?

**We thank the reviewer for this comment, as we should indeed discuss this. As can be seen in Fig. 3, the high MAR TOC during MIS 4 is linked to high sedimentation rates, whereas the peak in MAR TOC during MIS 5 is not. This would suggest that the MAR TOC maximum during MIS 5 actually resulted from increased primary productivity, whereas during MIS4 the high MAR TOC resulted from the increased sedimentation rate. This would in turn explain that there is no peak in *Chaetoceros* counts for this age. We have added the following (P. 14, lines 4-8)**
**"*During MIS 4 we also observe a peak in MAR$_{TOC}$ (Fig. 3a) but there is no corresponding peak in Chaetoceros abundance (Fig. 4c). However, the high MAR$_{TOC}$ during MIS 4 is linked to high sedimentation rates (Fig. 3c), whereas the peak in MAR$_{TOC}$ during MIS 5 is not, confirming that the MAR$_{TOC}$ maximum during MIS 5 indeed likely resulted from increased primary productivity, whereas during MIS 4 the high MAR$_{TOC}$ did not.*"**

P. 13, l. 17-18 & l. 25-26: since Proboscia is not a secondary component of diatom assemblages in coastal upwelling systems not it is not associated with high productive waters along the Chilean margin, these statements should be thoroughly revised. See my comments above for Introduction.

**We feel that the reviewer might have misunderstood our conclusions. As explained above, we argue that *Proboscia* is likely less abundant during intense upwelling in this region, and is therefore in fact not indicative of high productivity. In the introduction we explain that initially the Diol Index was proposed as an upwelling indicator since in general *Proboscia* is often associated with upwelling conditions. However, the actual conditions during/under which the species is abundant is often described as post-bloom, stratification, early upwelling season and/or the oceanic side of the upwelling front (e.g., Hart, 1942; Takahashi et al., 1994; Katsuki et al., 2003; Moita et al., 2003; Tarazona et al., 2003; Herrera and**

**Escribano, 2006; Sukhanova et al., 2006; see references in Table 1 of Rampen et al., 2014b), likely because *Proboscia* is only able to compete with other diatoms when silicate concentrations are low.**

P. 14, l. 2-5: This statement needs appropriate references/lab studies. Have different species of Proboscia been cultured the biomarker content measured in living cells?

**We have clarified that Sinninghe Damsté et al. (2003) cultured these *Proboscia* species and measured the lipid composition. Moreover, we have added the results of Rampen et al. (2007) who assessed the long-chain diol composition of *Proboscia inermis*, which also consisted for more than 90% of the $C_{28}$ 1,14-diol: "*In fact, Sinninghe Damsté et al. (2003) found that 98% of the diols P. alata consisted of the saturated and mono-unsaturated $C_{28}$ 1,14-diol, whereas P. indica produces similar amounts of the $C_{30}$ and $C_{28}$ 1,14-diol and P. inermis also mainly produced the $C_{28}$ 1,14-diol (Rampen et al., 2007).*"**

P. 14, l. 7: It is not correct stating that "P. alata needs little Si to build its frustule". For diatom standards, frustules of Proboscia are long and build long chains (see Jordan et al., 1991, Diatom Research 6, 63).

**Indeed, *Proboscia* diatoms have long chains but this is not the issue. Both Goering and Iverson (1981) and Sakka et al. (1999) suggest that *Proboscia* is capable of living under very low silicic acid concentrations because of weakly silicified frustules. Moreover, Jordan et al. (1991) state: "*The valves of modern Proboscia spp. are lightly silicified and thus their distribution in Antarctic sediments is restricted to regions of good preservation*". Thus, though the chains might be long, they are thin and weak.**

P. 14, 4.3. Sea surface temperature evolution: the discussion in this section jumps back and forth between different time windows. This is not reader-friendly. Revise

**To be honest, we do not really see how we jump back and forth between different time windows. However, we have divided the paragraph into smaller sections now and restructured the section a bit.**

P. 14, l. 22-25: Does your SST record following "global climate pattern" refers to MIS5 or the entire record? Please clarify

**We refer to the entire record, and we have clarified this.**

P. 15, l. 3: A correlation test helps to supports this statement.

**We have performed correlation tests and we plotted these in Fig. 6.**

P. 15, l. 7-8: This should be more rigorously discussed.

**We show here that our temperature proxies seem to accurately reflect past SST since the absolute difference between the proxy estimate are for most of the record within the maximal possible discrepancy that can be explained by the combined calibration errors. Further discussion of the proxy discrepancies is outside the scope of this paper in which we want to test the applicability of the diol proxies, and not that of the $U^{K'}_{37}$ and $TEX^{H}_{86}$.**

P. 15, l. 26-27: Looking at your Fig 7, several mismatches in the SST behavior of compared records are recognizable. This should be more critically and rigorously discussed (see for instance ODP1241 and GeoB3327-5 vs ODP1234).

5 **We merely wanted to show here that our $U^{K'}_{37}$ record overall agrees with other records in the vicinity of our site, but that in fact these records also show many discrepancies which is likely linked to the latitudinal movement of the ACC. For instance, site GeoB 3327-5 is located at the northern extent of the ACC, and thus largely influenced by its latitudinal movement (Ho et al., 2012), whereas for ODP 1234 this influence might be less. The $U^{K'}_{37}$ record of ODP 1241 is especially hard to compare with our record since this site is**
10 **low-latitudinal (6°N) whereas our site is located at 36°S, and thus the glacial-interglacial variability is much weaker at ODP 1241 as compared to our site. More extensive discussions are also a bit outside the scope of this paper as our main focus is to test the applicability of the proxies based on long-chain diols. We have added the following (P. 17, lines 9-14):**
*"Regional differences in timing of deglacial warming have previously been related to the southward shift of*
15 *the SWW, which directly and indirectly influences local paleoproductivity and upwelling intensity, and thereby potentially leaving site-dependent, unique signatures in the SST records (Mohtadi et al., 2008). This may partially explain the discrepancies that we observe between our $U^{K'}_{37}$ record and the other records. For instance, site GeoB 3327-5 is located at the northern extent of the ACC, and thus largely influenced by its latitudinal movement which is closely related to the migration of the SWW (Ho et al., 2012), whereas for ODP*
20 *1234 this influence might be less."*

The authors should comments and discussed on: - "The production/export depth of TEXH86 is not well constrained, thus complicating the comparison of TEXH and SST (for example, UK ) based records." (e.g., Kim
25 et al., 2012, EPSL 339, 95-102.; Ho & Laepple, 2018, Nat. Geosc. 9, 606).

**On page 16 we already mention that the TEX$_{86}$ might potentially reflect a subsurface signal, but that for this region it has been shown earlier that it likely reflects SST:** *"For the TEX$_{86}$, it has been shown to potentially reflect subsurface rather than surface water temperatures (Huguet et al., 2007; Kim et al., 2010;*
30 *2015; Schouten et al., 2013; Chen et al., 2014) due to the production of isoprenoid GDGTs below the surface mixed layer. Overall, the TEX$^{H}_{86}$ record agrees reasonably well with the $U^{K'}_{37}$ record for ODP 1234, suggesting that it mainly reflects SST. Also, Kaiser et al. (2015) who established a regional TEX$^{H}_{86}$ calibration suggested that this proxy mainly reflects SST."*

35 - "glacial–interglacial amplitude of TEXH86- derived SST change in the tropics is overestimated relative to other proxy evidence, a result also independently found by a multi-proxy study in the subpolar region" (Ho & Laepple, 2015, Earth Planet. Sci. Lett. 409, 15–22; Seki, O. et al. 2014. Prog. Oceanogr. 126, 254–266).

**We do not see why this is relevant as ODP 1234 is a subtropical site, and we do not see a larger TEX$^{H}_{86}$**
40 **glacial-interglacial amplitude as compared to the $U^{K'}_{37}$ and LDI. Furthermore, Zhang and Liu (2018) recently showed that core-top TEX$_{86}$ data between 30°N and 30°S strongly correlate to SST, which is in contrast with Ho and Laepple (2015) who proposed that TEX$_{86}$ data potentially reflect subsurface water temperatures which would cause the overestimation in glacial-interglacial SST change. Moreover, as our primary focus is to test the long-chain diol proxies we suggest that such a discussion is outside the scope of**
45 **this manuscript.**

**Response to Referee 2**

de Bar at al. test the applicability of different paleoenvironmental proxies based on long chain diols (LDI, Diol Index, and NDI ) by studying the ODP Site 1234 located within the Peru-Chile upwelling system and covering the last 150 kyrs. They compare LDI-derived SSTs with other temperature proxies (TEX$^H_{86}$, U$^{K'}_{37}$) and with the Diol index and NDI with other phytoplankton production proxies (accumulation rates of TOC and lipid biomarkers). Their results suggest that the Diol Index should not be considered as an upwelling proxy per se, and that the NDI might not be suitable as a more general paleonutrient proxy. I find this is an interesting study. I particularly appreciated the multi-proxy comparison for SST and productivity reconstructions. The overall manuscript is well structured and well written, even though some parts would need clarification. The data are robust and in general the conclusions are well supported by the data. I however think that some points in the discussion could be clarified/more detailed, as it is sometimes difficult to understand. Please find my comments below.

**We thank the referee for the positive assessment and for the comments, which we have used to improve the manuscript.**

P. 2, l. 12: "mean annual sea surface temperature" instead of "annual mean sea surface temperature".

**We have corrected this.**

P. 4, l. 16: throughout the text you use either "ka" or "kyr". I would be consistent and choose one or another.

**After suggestion of the editor, we keep this distinction.**

P. 9, l. 10: delete "average" as you talk about ranges.

**We have corrected this.**

P. 9, l. 13: it would be good to indicate the Terminations on the figures.

**We have indicated the Terminations in Figs. 3-5.**

P. 10, l. 8-9: the alkenone AR does not show this decrease around the boundary of MIS 4 and 5.

**Yes, this indeed true, and we have now added this. This difference between the concentration and AR is likely caused by the increase in sedimentation rate around the MIS 4-5 boundary.**

P. 12, l. 16-19: this sentence is not very clear and in contradiction. Please clarify.

**We agree that this sentence is not clear and seems contradictory; we have rephrased this sentence.**

P. 12, l. 19-32: this part is not really clear and relatively difficult to follow. Please rephrase.

**Referee 1 provided a similar comment. We have tried to describe this section more clearly.**

P. 13, l. 1: "individual lipid biomarkers" instead of "individual biomarker lipids".

**We have corrected this.**

P. 13, l. 1-12: How do you explain the peak of MARTOC in MIS 4 which does not correspond to a peak in loliolide concentration/Chaetoceros abundance? There is also no peak for the other lipid biomarkers, except for the 1,13 and 1,15-diols. Who are the potential producers of the 1,13 and 1,15-diols?

**We thank the reviewer for this comment, since this part indeed requires discussion; a similar comment was also given by reviewer #1.**
**Fig. 3 shows that the increase in MAR$_{TOC}$ in MIS 4 is concomitant with an increase in sedimentation rate, whereas during MIS 5 this is not the case. Therefore it is likely that whereas the peak in MAR$_{TOC}$ during MIS 5 is mainly caused by increased primary productivity (as also indicated by the other records), the MAR$_{TOC}$ maximum during MIS 4 is the result of enhanced particle settling due to the increase in sedimentation rate. Henceforth, we also do not observe any phytoplankton lipid biomarker peaks during this period.**
**The reason for the increase in both the concentration and AR of 1,13- and 1,15-diol around this period is unclear, as is the potential producers of these biomarkers. As explained in the Introduction, 1,13- and 1,15-diols have been observed in cultures of Eustigmatophyte algae, but these distributions do not agree with the distributions we observe in the marine realm, and thus the producer is still elusive.**
**We have added the following (P. 14, lines 4-8):**
**"*During MIS 4 we also observe a peak in MARTOC (Fig. 3a) but there is no corresponding peak in Chaetoceros abundance (Fig. 4c). However, the high MARTOC during MIS 4 is linked to high sedimentation rates (Fig. 3c), whereas the peak in MARTOC during MIS 5 is not, confirming that the MARTOC maximum during MIS 5 indeed likely resulted from increased primary productivity, whereas during MIS 4 the high MARTOC did not*"**

P. 15, l. 12-15: this sounds surprising as you state in P. 4, l. 2 that ODP 1234 "lies in the vicinity of two large Andean river systems". Has the influence of terrestrial input on diols been observed for recent times in this site?

**Yes, the core is located relatively close to the coast of Chile (~65 km) and the river mouths of the Río Bio-Bio and Río Itata, which both drain large basins. However, this distance is likely still too large to detect riverine influence in the organic matter deposited in the sediment. For instance, in the study of de Bar et al. (2016), the BIT index and the fractional abundance of the C$_{32}$ 1,15-diol was assessed in surface sediments along the river-influenced Iberian margin. High BIT and C$_{32}$ 1,15-diol values were only observed for stations which were located < 15 km from the coast.**

P. 15, l. 25-30: you could provide more details on the comparison with the other sites. Even though the general trend agrees well, there are clear differences with sites at proximity of ODP 1234 (GIK 17748-2; GeoB 3302-1).

**We thank reviewer #2 for this comment. Reviewer #1 has a comparable comment which we answer here in a similar way. We merely wanted to show here that our U$^{K'}_{37}$ record overall agrees with other records in**

the vicinity of our site, but that in fact these records also display many discrepancies that are likely linked to the latitudinal movement of the ACC. For instance, site GeoB 3327-5 is located at the northern extent of the ACC, and thus largely influenced by its latitudinal movement (Ho et al., 2012), whereas for ODP 1234 this influence might be less. Moreover, we believe that this is somewhat outside the scope of this paper as

5 our main focus is to test the applicability of the proxies based on long-chain diols. However, we have added the following (P. 17, lines 9-14):

*"Regional differences in timing of deglacial warming have previously been related to the southward shift of the SWW, which directly and indirectly influences local paleoproductivity and upwelling intensity, and thereby potentially leaving site-dependent, unique signatures in the SST records (Mohtadi et al., 2008). This may*

10 *partially explain the discrepancies that we observe between our $U^{K'}_{37}$ record and the other records. For instance, site GeoB 3327-5 is located at the northern extent of the ACC, and thus largely influenced by its latitudinal movement which is closely related to the migration of the SWW (Ho et al., 2012), whereas for ODP 1234 this influence might be less."*

P. 28, l. 8: a bracket is missing.

**We have corrected this.**

Figures 3, 4, 6, A1: the tick marks on the axes (especially the axis with the Age) are missing. Please add them so the reader can associate more easily the values with the data points.

**We have added these.**

**Response to Referee 3**

25 The authors of "A Late Quaternary climate record based on long chain diol proxies from the Chilean margin" analyzed a suite of organic biomarkers in a marine sediment core from the coast of Chile to a) assess glacial-interglacial dynamics in marine productivity (ie. upwelling) and associated climate change, and b) to establish the efficacy of long-chain diols in paleoenvironmental reconstruction, particularly with regards to the recently proposed Nutrient Diol Index (NDI) as a metric for marine productivity. They compared diol-based indices to

30 independent records of SST (alkenones and GDGTs) and productivity (algal lipid fluxes, total organic carbon fluxes, etc.) and show that NDI is distinctly different than other upwelling proxies and so is not widely interpretable as an upwelling indicator, suggesting instead that the principal C28 1,14-diol producer, Proboscia alata, occurs more commonly in non-upwelling settings. Furthermore, the Diol Index appears to reflect all Proboscia productivity, but still does not capture the upwelling signature expressed in other paleoproductivity

35 proxies. The dataset seems robust and these topics are of broad interest to the paleoceanography and organic geochemistry communities. In general, I think the paper could be published after addressing issues regarding the clarity of the paleoclimate implications and the writing structure.

**We thank the referee for the positive assessment and for the comments, which we discuss below and have used to improve our manuscript.**

Please include the full name for each abbreviation the first time abbreviations are mentioned. Likewise, provide the diol-specific equations in the introduction because it would help the reader follow the introduction section easier without flipping through the paper to find the equations.

**We have included the full names. However, we do not think that including the diol equations in the introduction will improve the clarity of the text, and we already describe specifically on which LCDs the equations are based.**

The goal stated in the first sentence of the abstract stands in contrast with the goal of the title.

**We agree with the referee, and we have changed the first sentence of the abstract.**

Page 12-13: The relationship between temperature and productivity should be discussed more explicitly. I think this section relating the broader climate system and the productivity regime needs clarification. A plot showing the temperature and the nutrient/upwelling indicators together is warranted, given that the paper is drawing links between these two variables. Can either 'northward migration of the SWW' or 'southward migration of the subtropical high' or global cooling induce more upwelling and is there a way to distinguish between these factors in the paleoceanographic record?

**We are not sure on how to more explicitly discuss the relation between productivity and temperature. We start the discussion by illustrating the general idea on how the latitudinal movement of the ACC and SWW linked to the transition of the Last Glacial Maximum to the Holocene might affect productivity. In short, different studies suggest that during the LGM productivity was likely stimulated by greater nutrient input deriving from the ACC (being in the north) and enhanced river runoff due to the northern position of the SWW. Accordingly, productivity would be lower during the Holocene due to the southern position of the ACC and SWW. However, during the Holocene, the winds associated with the subtropical high pressure system might induce upwelling. Moreover, there are also studies that propose that the northern position of the SWW during the LGM might in fact prevent upwelling and thus inhibit productivity despite the greater input of nutrients. Hence, we discuss coastal productivity in light of the main current and wind regimes linked to glacial-interglacial variability. However, most studies have focused on the last ~30 kyrs, and therefore little is still known about upwelling and productivity during for instance the Last Interglacial. Nevertheless, we discuss our biomarker records in light of the knowledge that we have of the LGM and Holocene.**

Page 14: Many ideas are put forward about P. alata ecology, but I am left with an uncertain idea of what NDI (aka P. alata productivity) variations signify in a specific climatic or ecological sense. Is there a suggestion of why P. alata varies so dramatically during some periods, but not during MIS 1/2/3/4? I think the discussion on the modern observations of P. alata blooms and distributions are helpful, but should be related more explicitly to the paleoceanographic record if possible. Is the author's suggestion that reduced NDI around 100 ka is an indication that upwelling occurred continuously, rather than on a seasonal basis, thereby eliminating the "early- or post-upwelling nutrient conditions"?

**Both the NDI and the Diol Index reflect the relative abundance of *Proboscia* diatoms as the 1,14-diols are specific for this group of diatoms. Since along the Chilean margin *Proboscia alata* has been observed, we propose that for our location the NDI reflects *P. alata* productivity. However, in other regions the $C_{28}$ 1,14-**

**diol** might be produced by other *Proboscia* species and the NDI might thus not reflect solely *P. alata* productivity. In this light we have also partly adjusted our hypothesis that the NDI potentially reflects *P. alata* productivity; it is merely more species-specific as compared to the Diol Index since it excludes the $C_{30}$ 1,14-diol from the numerator. We have added the following (P. 15, lines 16-18):

5 "*In other regions, the $C_{28}$ 1,14-diol might be produced by other Proboscia species (e.g, P. inermis) and the NDI might thus not reflect solely P. alata productivity. In summary, we suggest that the NDI at the Chilean margin likely reflects Proboscia productivity, and may therefore not be suitable as paleo-nutrient tracer.*"

As for the question why the NDI/*Proboscia* activity is relatively constant during MIS 1/2/3/4 but shows
10 large variation during MIS 5: this is one of the main issues we have addressed in the manuscript. We propose, based on the maximum in $MAR_{TOC}$ during MIS 5 concurrent with e.g. the highest abundances of preserved *Chaetoceros* valves, that around 100 ka the upwelling intensity was likely to be strong and primary productivity was stimulated. What we know from the modern-day *Proboscia* ecology, is that this diatom genus proliferates when nutrients increase, i.e. during upwelling. However, it is particularly able of
15 dominating the diatom community during early upwelling conditions when silicate concentrations are still low, and the nutrients are still in the deeper waters. When actual upwelling occurs, and the nutrients are transported to the surface and silicate concentrations increase, *Proboscia* is likely outcompeted by for instance *Chaetoceros* which is more heavily silicified. Accordingly, we think this might explain the relatively low Diol Index and NDI around 100 ka, as the relative abundance of *Proboscia* was likely to be
20 low. However, before and after this event, i.e. around ca. 120 and 90 ka, upwelling was likely less intense, and conditions were more favorable for *Proboscia*.
As for the referee's last question: the reduced NDI around 100 ka does not indicate that upwelling occurred continuously throughout the year and thus does not eliminate the seasonal basis of upwelling. However, the NDI and Diol Index reflect integrations of multiple years and thus an averaged signal of
25 *Proboscia* productivity over these years. We believe that the minimum in the NDI around 100 ka, suggesting that phosphate and nitrate concentrations were around 0 µmol/L, is not realistic, because the "mean annual nutrient concentrations" were not likely to be ~0 during this upwelling interval. Even though nutrients can be used up at the end of an upwelling season, mean annual nutrient concentrations are likely to be relatively high.

30 There is an increase in TOC-MAR and importantly, 1,13- and 1,15-diols in MIS 4. Please comment on if this is an indication of specific upwelling or nutrient conditions that might promote the eustigmatophyte but not other algal groups.

Yes, this is true, and we have included this in the revised manuscript. Moreover, similar comments were put forward by referees 1 and 2. We believe that the increase in $MAR_{TOC}$ during MIS 4 is likely linked to
35 the evident increase in sedimentation rate during this period (Fig. 3a). During MIS 5, the maximum in $MAR_{TOC}$ corresponds to elevated TOC levels and a constant sedimentation rate, suggesting an increase in primary productivity. However, during MIS 4, the TOC levels are almost twice as low, but the sedimentation rate is clearly higher, suggesting that the increased $MAR_{TOC}$ is a result of enhanced particle settling. We have added the following:
40 "*During MIS 4 we also observe a peak in $MAR_{TOC}$ (Fig. 3a) but there is no corresponding peak in Chaetoceros abundance (Fig. 4c). However, the high $MAR_{TOC}$ during MIS 4 is linked to high sedimentation rates (Fig. 3c),*

*whereas the peak in MAR$_{TOC}$ during MIS 5 is not, confirming that the MAR$_{TOC}$ maximum during MIS 5 indeed likely resulted from increased primary productivity, whereas during MIS 4 the high MAR$_{TOC}$ did not"*

P. 15 Starting line 31: Regional differences in the timing of deglacial warming might be expected, but this does not address the different timing among proxies at Site 1234. Please elaborate on the significance of the LDI versus UK37 difference during ∼10 kyr around the LGM.

**Yes, we completely agree with referee 3, however, we are not certain on how to explain this ±10 kyr time lag between the deglacial warming as recorded by the LDI and the $U^{K´}_{37}$ and $TEX^H_{86}$. The LDI-temperatures are also considerably lower as compared to the other proxy temperatures. Although the producers of the 1,13- and 1,15-diols are unknown, they are likely to be phototrophs living in the upper part of the photic zone (e.g., Rampen et al., 2012; Balzano et al., 2018), and therefore we would expect similar temperature estimates for the LDI and $U^{K´}_{37}$. We have added the following (P. 16, lines 24-27) "*Potentially, the production of the 1,13- and 1,15-diols shifted to a colder season due to regional climatic change associated with the southward migration of the ACC and SWW upon the deglacial warming. Finally, wo observe an early onset of the deglacial warming for the LDI compared to the $U^{K´}_{37}$ and $TEX^H_{86}$ (ca. 6 kyrs; Fig. 5b), the reasons for which are elusive*"**

Some points need grammatical correction:

P1. L. 11: Change "proxies" to "indices" because the Diol Index is stated to be an indicator rather than a proxy in Line 12.

**We have adjusted this.**

P1. L.12: ". . .the NDI as a quantitative proxy. . ."

**We have rephrased as follows: "*… the NDI index as proxy for…*".**

P1. L. 15: Either specify the number of glacial/interglacial periods or just remove this clause.

**We have removed this.**

P2. L. 16:: Provide the equation for DI, or rephrase, because it is not simply the ratio, but rather the percentage.

**We agree that this formulation is unclear, and we have rephrased as follows (P. 2, lines 16-17): "*The Diol Index, which is an indicator for past upwelling/high nutrient conditions, is defined as the fractional abundance of 1,14-diols with respect to 1,13-diols (Willmott et al., 2010) or the C$_{30}$ 1,15-diol (Rampen et al., 2008)…*"**

P2. L 24: Remove "sea", or change to "marine"?

**We have changed this into "marine".**

P2. L 24-25: If possible, specify if the change in saturation is related to changing species or interspecies diol adjustments.

**This is not known. In sediments, mono-unsaturated 1,14-diols are generally lower in abundance as compared to the saturated 1,14-diols. However, culture data show that *Proboscia* diatoms often produce more unsaturated diols, and that the degree of unsaturation is strongly affected by temperature (Rampen et al., 2009; OG). This temperature relationship was however not observed in marine surface sediments (Rampen et al., 2014) and the low abundance of unsaturated 1,14-diols might be the result of preferential degradation. However, the regional differences in 1,14-diol distributions are likely also the result of different species producing different 1,14-diols.**

P2. L. 27: Remove "Preliminary"

**We have removed this.**

P2. L28: "paleo-upwelling" could be used instead of past upwelling. Remove "as". Also, change "proxies" to "indicators" unless the reconstruction of upwelling rates is a quantitative transfer function.

**We have corrected this.**

P3. L. 10: The Holocene is part of the Quaternary – no need to mention separately. orzet

**We have removed "and Holocene".**

P3. L. 10: "large climate cycle" is kind of arbitrary

**We have removed "covering the last large climate cycle of the Pleistocene"**

P3. L. 10: remove "bulk organic matter"

**We will have removed this.**

P3. L. 29: Can you include more specifics about the connection – namely, specify if stronger ACC and Westerlies result in warmer or cooler climate.

**The relationship between the ACC and Westerlies with productivity and upwelling is described in section 4.1, but we agree that we should elaborate more on the link between the ACC/Westerlies and temperature, which we have done as follows (P. 3 line 32 – P. 4 line 4):**
**"*Lamy et al. (2002) showed that long-term Holocene trends in temperature and productivity are linked to latitudinal shifts of the ACC and the Westerlies. On a millennial- to multidecennial-scale, paleotemperature variations are strongly connected to millennial-scale climate variations in Antarctica. When the ACC migrates northward, potentially associated with an expansion of Antarctic sea ice, cold, subantarctic waters are advected into the southern hemisphere midlatitudes, including the Chilean margin (e.g., Lamy et al., 2004; Kaiser et al., 2005).*"**

P. 3 L. 30-33. Does not follow from previous part of paragraph. I think it needs a segue from climate to upwelling/sedimentation dynamics, because this statement isn't about climate like the rest of paragraph. Also, please describe the subtropical high pressure system clearly because it is referred to later as a possible explanation.

**We agree that this structure was not optimal, and we have moved these lines to the second part of the paragraph (P. 4, lines 11-14). We have now included one sentence on the subtropical high-pressure system in the 'Study site' section. However, we have now removed the sentence in which we suggest that the subtropical high-pressure system might potentially be related to the upwelling event during the Last Interglacial, since this is rather speculative.**

P. 4 L. 2: There might be some variation in this, but I think a better spelling is Biobío

**We have corrected this.**

P. 4, L. 3: remove "both" and "large" to read ". . .which drain basins of 24,000 km2. . ."

**We have corrected this.**

P. 4 L. 13: Remove "providing the potential for high-resolution records" because it's not necessary and I don't think the author's did that kind of sampling.

**We have removed this.**

P. 5 L. 5: If these benthic foraminifera data are already published, I don't think this paragraph is necessary, is it? Are any adjustments made to the McManus and Heusser paper? For brevity, this could be folded into the age model paragraph because it provides the basis for the correlations of the cores. This change is not critical for publishing the paper.

**The correction of 0.64‰ is new information and therefore relevant. Since the planktonics are described in this paragraph, we prefer to keep the information on the benthic data also in this section.**

P. 6 L 9: Specify that the GDGT naming scheme in the following descriptions relates to the structures in Castañeda and Schouten, or some other appropriate reference

**We refer to Sinninghe Damsté et al. (2002).**

P. 6 L. 28: Remove "in this case" and clarify that the BAYSPAR refers to the Tierney and Tingley citation.

**We believe that "in this case" is relevant here, since the core-tops on which the calibration is based, is different for every paleo-reconstruction; it depends on the assumed prior mean temperature and search tolerance. We used a prior mean of 13 °C and a search tolerance of 0.2 TEX$_{86}$ units, resulting in the 62 high latitudinal grid boxes on which in this case the calibration is based. We now refer to Tierney and Tingley (2014, 2015).**

P. 6 L. 31-32. Awkward phrasing because of passive voice.

**We have rephrased this sentence in present sense.**

P. 7. "Conductive"?

**We thank the referee for remarking this typo: it should be "conducive".**

P. 7, L. 7: "isomers can indicate these types of environments in the past. . ." should be rephrased because it is confusing as written.

**We have rephrased this.**

P. 7 L. 8: Include a reference suggesting 0.3 is acceptable threshold for Methane Index.

**We refer here to Zhang et al. (2011):** *"Combined GDGT distribution and isotopic evidence suggest that 0.3 to 0.5 of the MI might be a reasonable threshold for distinguishing gas hydrate impacted and/or methane-rich environments from normal marine realm."*

P. 7 L 34: What was the injector configuration?
**On-column injection, we have now clarified this.**

P. 8 L. 8: Helium does not need to be capitalized.
**We have corrected this.**

P. 8 L. 23: The first part of this sentence should be rephrased because it is confusing as written.
**We have rephrased this.**

P. 8 L. 27: "..potentially riverine derived. . ." should be rephrased for grammar.
**We have rephrased as follows: "***which is potentially derived from rivers***".**

P. 8 L. 30: Given that you are testing the effectiveness of the NDI index, insert: "..The NDI index, a proposed proxy for PO43-. . ."
**We have corrected this.**

P. 10 L 34: Insert a comma after "In all sediments"
**We have corrected this.**

P. 10: It seems strange that crenarchaeol is the first biomarker presented when it has only been mentioned as a standard in the first part of the paper, and is never discussed or interpreted in any details. Could remove statement about crenarchaeol, else it should be discussed. Similarly, figures 4f and 4g and 4h come before 4a, 4b, etc. in the text. I recommend rearranging this so that diol results come first and things don't seem out of order, or just refer to them as figure 4.
**We have clarified that crenarchaeol is a specific biomarker for Thaumarchaeota (P. 10, line 18):**
**We agree that the order in Figure 4 is in contrast with the results, and we have rearranged the panels in this figure, as well as some paragraphs in the methods section.**

P 10 L 4: The reported accumulation rate for crenarchaeol does not match what is shown in figure 4f.

**We thank the referee for noticing this; we have corrected this.**

P. 10 L. 16: Figure "4i", not figure 4j. Also, the data are labeled pg g-1, but the text states µg g-1. Likewise ng versus mg for the 1,14-diol accumulation rates.

**We thank the referee for pointing out these inaccuracies, and we have corrected these.**

P. 11 L 26-27: State on which proxies these estimates are based.

**We agree that this was unclear, and we have clarified that this is based on all three proxies.**

P. 11, L 28: The spline curves show a decrease of 4-6∘C not 6-7∘C. Smallest change is for iGDGTs.

**We have corrected this, as all proxies show a ca. 4-6C temperature decrease.**

P. 12 L 27: ". . .since neither opal concentration nor opal or TOC-MAR increased simultaneously with TOC concentration" might help clarify this sentence.

**We have changed accordingly.**

P. 13, L 16: Intervals of enhanced upwelling are not explicitly labeled in Figs 3b and 3c.

**We agree that this is unclear and we have changed this as follows: "… *with enhanced upwelling (i.e., around 100 ka; purple band in Fig. 3)*".**

P. 13, L 27. Rephrase "As for the diol index and the. . ." because this phrase has another common meaning than as it is used.

**We have rephrased this.**

P. 14 L 10: "during/under" could just be "in"

**We have corrected this.**

Figure 4: How can there be sometimes more chaetoceros counts than total diatom counts?

**This is because the counts are not quantitative, but descriptive: F=few; C=common; A=abundant; VA=very abundant. However, we now realize that we did not clarify this is in the caption, which we have now corrected.**

Figure 5: What are the orange versus blue lines in the Site 1234 Benthic oxygen isotopes? It is not mentioned in the figure caption.

**We thank the referee for noticing this, and we have now clarified this in the caption. The color scheme is similar to Fig. 2, where the two colors reflect the two separate age models. 
[revised manuscript text omitted]